# Sub-mesoscale observations of convective cold pools with a dense station network in Hamburg, Germany

Bastian Kirsch[1,4], Cathy Hohenegger[2,4], Daniel Klocke[3,4,2], Rainer Senke[1], Michael Offermann[1], and Felix Ament[1,2,4]

[1]Meteorological Institute, University of Hamburg, Germany
[2]Max Planck Institute for Meteorology, Hamburg, Germany
[3]Deutscher Wetterdienst, Offenbach am Main, Germany
[4]Hans Ertel Centre for Weather Research, Branch Model Development - Convection, Hamburg, Germany

**Correspondence:** Bastian Kirsch (bastian.kirsch@uni-hamburg.de)

**Abstract.** From June to August 2020 an observational network of 103 meteorological ground-based stations covered the greater area ($50\,\mathrm{km} \times 35\,\mathrm{km}$) of Hamburg (Germany) as part of the Field Experiment on Sub-mesoscale Spatio-Temporal variability at Hanseatic city of Hamburg (FESST@HH). The purpose of the experiment was to shed light on the sub-mesoscale ($\mathcal{O}(100)\,\mathrm{m} - \mathcal{O}(10)\,\mathrm{km}$) structure of convective cold pools that typically remains under-resolved in operational networks. During the experiment, 82 custom-built, low-cost APOLLO (Autonomous cold POoL LOgger) stations sampled air temperature and pressure with fast-response sensors at $1\,\mathrm{s}$ resolution to adequately capture the strong and rapid perturbations associated with propagating cold pool fronts. A secondary network of 21 weather stations with commercial sensors provided additional information on relative humidity, wind speed and precipitation at $10\,\mathrm{s}$ resolution. The realization of the experiment during the COVID-19 pandemic was facilitated by a large number of volunteers who provided measurement sites on their premises and supported station maintenance. This article introduces the novel type of autonomously operating instruments, their measurement characteristics, and the FESST@HH data set (https://doi.org/10.25592/UHHFDM.10172; Kirsch et al., 2021b). A case study demonstrates that the network is capable of mapping the horizontal structure of the temperature signal inside a cold pool, as well as quantifying a cold pool's size and propagation velocity throughout its life cycle. Beyond its primary purpose, the data set offers new insights into the spatial and temporal characteristics of the nocturnal urban heat island and variations of turbulent temperature fluctuations associated with different urban and natural environments.

## 1 Introduction

We present the instrumentation and data set of the Field Experiment on Sub-mesoscale Spatio-Temporal variability at Hanseatic city of Hamburg (FESST@HH) that illuminates the spatial and temporal structure of convective cold pools. Conventional mesoscale meteorological station networks have a typical resolution of $\sim 25\,\mathrm{km}$ in space and $10\,\mathrm{min}$ in time and are essential in order to obtain a reliable picture of the atmospheric background state. Remote sensing techniques can help us to retrieve more-finely resolved information for a few atmospheric variables like precipitation and wind speed. However, the spatial variability of basic meteorological parameters like air temperature or pressure on the sub-mesoscale (i.e., length scales between $\mathcal{O}(100)\,\mathrm{m}$

and $\mathcal{O}(10)\,\mathrm{km}$) remains under-resolved. Information on sub-mesoscale variability is especially important for the investigation and simulation of convective clouds and precipitation (Stevens et al., 2020). Convective cold pools are an important source of this variability and represent a blind spot for conventional observation networks. Therefore, cold pools are the main motivation for the FESST@HH experiment.

Cold pools play a prominent role for understanding the formation and life cycle of atmospheric convection. They are defined as areas of relatively cool and dense air that form through melting and evaporation of hydrometeors underneath precipitating clouds. Their horizontal size range from a few $\mathrm{km}$ to hundreds of $\mathrm{km}$ (Terai and Wood, 2013; Feng et al., 2015; Zuidema et al., 2017) and temperature decreases can exceed $10\,\mathrm{K}$ (Engerer et al., 2008; Kirsch et al., 2021a). As the body of cold air grows and propagates horizontally away from the precipitation, it often causes a rapid decrease in local air temperature and the formation of a gust front. Secondary updrafts are preferentially triggered in the region ahead of the cold pool air due to the combination of mechanical lifting and accumulation of moisture in the lower troposphere (Tompkins, 2001; Feng et al., 2015; Torri et al., 2015; Drager et al., 2020). Cold pools are also found to support the organization of convective clouds and the transition from shallow to deep convection (e. g., Rotunno et al., 1988; Khairoutdinov and Randall, 2006; Schlemmer and Hohenegger, 2014; Kurowski et al., 2018). Since models with hectometer-scale resolution are required to realistically represent processes related to convection (Bryan et al., 2003; Grant and van den Heever, 2016; Hirt et al., 2020), large-eddy simulations (LES) have become an established tool to study cold pools. However, LES models critically depend on parametrizations of subgrid-scale phenomena, such as turbulence and microphysics (Smagorinsky, 1963; Dawson et al., 2010; Li et al., 2015). This fact supports the need for observational data with a similar resolution to LES models that facilitate validation of simulations and enhance our understanding of sub-mesoscale processes.

Observational studies on cold pools draw upon a wide range of data-collection methods. These include measurements from boundary layer towers (Goff, 1976; Kirsch et al., 2021a), aircraft (Terai and Wood, 2013), ships (de Szoeke et al., 2017), precipitation radars (Borque et al., 2020), and combinations thereof (Mueller and Carbone, 1987; Feng et al., 2015). Nevertheless, capturing the horizontal structure of cold pools with the help of in situ observations is a challenging task given the small number of studies using, e. g., mesoscale station networks (Markowski et al., 2002; Engerer et al., 2008). In a recent study, van den Heever et al. (2021) described the use of uncrewed aerial systems, radiosondes and three closely-spaced surface stations during the C$^3$LOUD-Ex campaign to characterize the structure of convective updrafts and cold pools. The observations revealed that cold pool temperatures can exhibit variability on spatial scales between $\mathcal{O}(100)\,\mathrm{m}$ and $\mathcal{O}(1)\,\mathrm{km}$. However, given the small geographical extent of the in-situ network, the experiment was unable to characterize the complete surface temperature structure of any individual cold pool. There are also examples of more extensive observation networks at $\mathrm{km}$-scale resolution, like the Oklahoma City Micronet (Basara et al., 2011) and the WegenerNet (Kirchengast et al., 2014). However, with respect to areal coverage and instrument characteristics, these networks were mainly designed for climate monitoring rather than investigating short-lived convective-scale phenomena like cold pools.

The FESST@HH field experiment took place in Hamburg (Germany) from June to August 2020. The aim of the experiment was to perform meteorological observations that are suited to capturing the spatial structure of cold pools at the sub-mesoscale. This goal required the design of a ground-based station network that is large enough to cover the typical size of a cold pool

($\mathcal{O}(10)\,\mathrm{km}$) and dense enough to satisfy the desired resolution ($\mathcal{O}(100)\,\mathrm{m}$). Additionally, the measurement instruments had to be equipped with appropriate sensors for capturing cold pool fronts but still inexpensive enough to be deployed in large numbers. The relatively long measurement period was chosen to catch a reasonable number of cold pools during the convective season in Hamburg, where around seven events per month are expected (Kirsch et al., 2021a). In this article, we introduce the novel instruments and data set of FESST@HH and demonstrate its potential for investigating the spatial structure and life cycle of cold pools based on a case study. Moreover, the experimental setup allows for the assessment of the spatial dimension of the nocturnal urban heat island as well as the variability of turbulent temperature fluctuations in different environments. A special feature of FESST@HH was the participation of a large number of volunteers that became necessary due to the COVID-19 pandemic.

## 2 Design of instruments

The instrument design for the FESST@HH experiment was guided by the technical demands required to observe the sub-mesoscale structure of cold pools in space and time. Most importantly, the instruments had to be equipped with fast-response air temperature sensors that precisely capture relative changes in temperature. The sampling interval required was $\Delta t \leq 10\,\mathrm{s}$ in order to track the propagation of cold pool fronts with a velocity of about $10\,\mathrm{m\,s^{-1}}$ (Borque et al., 2020) on scales of $\mathcal{O}(100)\,\mathrm{m}$. Moreover, the instruments had to be able to operate independently of external power for about two weeks given site location constraints. Based on these requirements, we designed and manufactured the Autonomous cold POoL LOgger (APOLLO) for operation during the FESST@HH experiment (section 2.1). We complemented the network with WXT weather stations (section 2.2) based on commercial sensors.

### 2.1 APOLLO stations

The APOLLO is a simple and low-cost data logger that records air temperature and pressure. Inspired by the pyranometer network stations used during the HOPE campaign (Madhavan et al., 2016), it operates autonomously without an external power source, and its sensor and control units are mounted on a short mast (Fig. 1). The APOLLO is equipped with a fast-response and moisture-resistant thermometer consisting of a small ($7\,\mathrm{mm} \times 1.2\,\mathrm{mm}$) NTC (negative temperature coefficient) thermistor probe (type: TE Connectivity GA10K3MRBD1; Fig. 2a) placed inside a passively ventilated radiation shield (see sensor specifications in Table 1). The control unit is based on a micro controller board (type: HIMALAYA Matrix-Core ESP32) located inside a small ($26\,\mathrm{cm} \times 17\,\mathrm{cm} \times 10\,\mathrm{cm}$) logger box (Fig. 2b). A digital air pressure sensor (type: Bosch BME280 Environmental sensor) is installed on the logger main board, and a counterbalance valve ensures rapid adjustment of barometric pressure inside the logger box.

The readings of the temperature and pressure sensors are sampled with a resolution of $1\,\mathrm{s}$ and recorded onto a micro-SD memory card. An on-board GPS receiver is used to initially synchronize the internal real-time clock on start of the logger. To minimize power consumption during operation, the GPS module is activated to log the current time only once per hour for about $10\,\mathrm{s}$, which allows correction of the logger time stamp for drift in a post-processing step. The logger can establish a local

WiFi hotspot for on-site control, data inspection and fast data access via download and is able to send status messages via the LoRa (Long Range) wireless communication protocol for live monitoring.

The APOLLO is powered by a standard USB power bank battery ($20\,\mathrm{Ah}$ capacity) that corresponds to an autonomous operation time of 10 to 14 days. For easy access during maintenance, the power bank is placed inside a separate plastic tube. All logger units are mounted on a $3\,\mathrm{m}$ mast that is anchored in the soil with a ground screw. In the standard configuration, the temperature sensor at the top of the mast and the pressure sensor inside the logger box are situated about $2.9\,\mathrm{m}$ and $1.7\,\mathrm{m}$ above ground, respectively. The material costs for all required parts sum up to around 300 Euro per station.

The APOLLO station prove to be especially suited for accurately recording sharp thermal boundaries associated with cold pool front passages. In wind tunnel laboratory tests the shielded NTC temperature sensor exhibits a response time (i.e., $e$-folding time constant) of $\tau = \{93, 14, 10\}\,\mathrm{s} \pm 10\,\%$ at $\{1, 3, 5\}\,\mathrm{m\,s^{-1}}$ wind speed when being artificially heated and released to room temperature. Since cold pools are usually associated with wind speed increases of $3\,\mathrm{m\,s^{-1}}$ and more (Kirsch et al., 2021a), the lack of active sensor ventilation is unlikely to impact the accuracy of temperature measurements of cold pool passages. The comparison with an ultrasonic sensor confirms that the NTC thermometer captures well the rapid cooling signature of a cold pool event (Fig. 3). Although the response time of the APOLLO sensor is too large to respond to sub-second temperature fluctuations in the same way as the inertia-free ultrasonic sensor, it does not show any apparent lag in the minute-scale shape of the signal and even captures second-scale fluctuations. Moreover, trial field measurements of two cold pool events show that external site conditions, such as surrounding obstacles or different surface properties, do not systematically impact the strength and shape of a cold pool temperature signal (not shown) and, therefore, do not restrict the choice of measurement sites for APOLLO stations. Since this is not true for observations of wind speed and wind direction, these are only performed by the WXT weather stations at appropriate locations. Similarly, humidity measurements are limited to the WXT stations to avoid the technical and financial effort needed to equip the APOLLO stations with precise sensors.

## 2.2 WXT weather stations

WXT weather stations are employed to complement the APOLLO stations by providing information on other common meteorological parameters at selected locations. The main component of the station is a commercial compact multi-parameter sensor (type: Vaisala Weather Transmitter WXT536) that measures air pressure, temperature, relative humidity, wind speed, wind direction, and precipitation (see sensor specifications in Table 2). Pressure, temperature and relative humidity measurements are performed by a PTU module inside a radiation shield that combines a capacitive silicon BAROCAP sensor (pressure), a resistive thin-film Pt1000 sensor (temperature) and a capacitive thin-film polymer HUMICAP R2 sensor (relative humidity). Wind speed and wind direction are detected by an WINDCAP ultrasonic anemometer on top of the radiation shield that consists of three equally-spaced ultrasonic transducers on a horizontal plane. The wind measurements are determined from the transit times of the ultrasound along the three paths with a sampling rate of $4\,\mathrm{Hz}$ and are internally averaged. Wind gusts are defined as the highest average wind speed of a $3\,\mathrm{s}$ interval which is internally updated every second. The precipitation measurement principle of the WXT536 sensor is based on a RAINCAP piezoelectrical sensor underneath a steel cover that detects the impacts of individual rain drops within a collecting area of $60\,\mathrm{cm^2}$. Since the individual signals are proportional to the size

of the rain drops, the sensor is able to derive the accumulated rainfall amount within the measurement interval. The sensor also distinguishes between impacts of rain drops and hail stones (Vaisala, 2020). Additionally, the WXT station is equipped with an external Pt1000 thermometer (type: TMH cable sensor Pt1000 1/3 DIN Klasse B; size: $40\,\text{mm} \times 3\,\text{mm}$) inside a separate radiation shield that allows for smaller response times of the measured temperature signal ($\tau = \{173, 55, 39\}\,\text{s} \pm 10\,\%$ at $\{1, 3, 5\}\,\text{m}\,\text{s}^{-1}$ wind speed) than the internal PTU module ($\tau = \{334, 213, 155\}\,\text{s} \pm 10\,\%$ at $\{1, 3, 5\}\,\text{m}\,\text{s}^{-1}$ wind speed).

The WXT data are sampled at $10\,\text{s}$ resolution and written onto a SD memory card by a data logger (type: Avisaro M22766) that is synchronized with an integrated GPS module. Each station is powered by a $12\,\text{V}$ lead battery. The battery is recharged by a solar panel ($67\,\text{cm} \times 41\,\text{cm}$) so that the station can operate autonomously for several months, depending on the available sunlight. Similar to the APOLLO design, the station components are mounted on a $3\,\text{m}$ mast that is anchored in the soil with a ground screw (Fig. 4).

Similar to the APOLLO stations, site conditions like surrounding obstacles and surface properties are not expected to have a systematic impact on cold pool signals in temperature, pressure and humidity that is specific to the WXT sensors. However, wind speed and wind direction are heavily affected by nearby obstacles like buildings or vegetation due to the low installation height of $3\,\text{m}$ above ground. Therefore, the individual site properties have to be taken into account when choosing a measurement location as well as interpreting the wind data of the WXT stations.

## 3 Description of experiment

The FESST@HH measurement campaign took place under extraordinary circumstances. Its name is a modification of the acronym FESSTVaL (Field Experiment on Sub-mesoscale Spatio-Temporal Variability in Lindenberg), a field campaign that was originally planned to take place at the Meteorological Observatory Lindenberg (eastern Germany) during summer 2020. Due to travel restrictions associated with the COVID-19 pandemic, the decision was made to postpone the FESSTVaL campaign to 2021 and to replicate the cold pool part of FESSTVaL in Hamburg under the name FESST@HH to make it compatible with pandemic-related regulations.

### 3.1 Experiment area

Hamburg is the second largest city in Germany (population: 1.9 million; 2019) and is located in northern Germany ($53.5°$ N, $10.0°$ E) about $80\,\text{km}$ from the North Sea and the Baltic Sea. The FESST@HH measurement sites cover an area of $50\,\text{km} \times 35\,\text{km}$ that includes the urban center of Hamburg and its rural surroundings (Fig. 5). The station network consists of 82 APOLLO sites (Table 3) and 21 WXT sites (Table 4) that are non-uniformly scattered over the domain with a generally higher station density closer to the city center. The arrangement of stations results from the location of private gardens and of public facilities like schoolyards that could be used as measurement sites (see section 3.2). Moreover, the sites for WXT stations were chosen based on a small impact of surrounding obstacles on the local wind field and a roughly uniform distribution over the network. The average nearest-neighbor distance between all 103 measurement sites is $1.85\,\text{km}$ with a standard deviation of $1.42\,\text{km}$, whereas it is $1.93 \pm 1.41\,\text{km}$ and $5.49 \pm 1.98\,\text{km}$ for APOLLO and WXT, respectively. The measurement area is crossed by

the Elbe river in southeast–northwest direction and is characterized by relatively flat terrain. While the Elbe valley is situated around sea level, the elevation north and south of it does not exceed about 80 m and 110 m, respectively. The altitudes of all measurement site lie between 1 m and 82 m above sea level with an average of 17 m.

To characterize the environmental properties of the measurement sites, we apply the local climate zone (LCZ) framework introduced by Stewart and Oke (2012). This framework classifies the impacts of surface structure, surface cover and human activity on the local thermal climate with the help of 17 standardized LCZ classes. All 103 FESST@HH sites fall into 15 different LCZ classes, which also includes combinations of classes to describe the heterogeneity within about 500 m around the station (Bechtel et al., 2015). The majority of sites (61) are situated in an open arrangement of low-rise buildings and scattered trees (LCZ 6; *mixed* in Fig. 5), while 24 sites, mostly near the city center, are located in the proximity of mid-rise and high-rise buildings, either in a compact or open arrangement (LCZ 2, 4 and 5; *urban*). Each of the other occurring classes contain less than ten sites, which include natural environments (LCZ 9, A, B and D) and industrial areas with mostly paved surfaces (LCZ 8 and 10). Ten sites are situated less than 50 m away from water bodies like the Elbe river, the Alster lake, or channels (LCZ G).

## 3.2 Implementation

The measurement period started on 1 June 2020 and ended on 31 August 2020, whereas the the installation of all stations was completed in mid-June. The realization of the experiment was enabled by the support of many institutions and private landowners who provided permission at short notice to use their premises as measurement sites. The different groups of landowners are indicated by two-letter acronyms as part of a unique site identifier code that also contains the site number and the installed instrument type (a=APOLLO, w=WXT). The majority of measurement stations were installed in private gardens and backyards (PG; Fig. 6a) as well as on the grounds of various institutions and clubs (OG). Further groups of sites include existing weather stations and sites of the Meteorological Institute (WS and MI), air quality observation sites of the Free and Hanseatic City of Hamburg (LM), grounds of the University of Hamburg (UH), and schoolyards (KB). A special characteristic of the latter group is that most of the respective APOLLO logger components are mounted at existing weather stations used for educational purposes and, therefore, exhibit lower temperature sensor heights (∼1.8 m) than the standard stations (Fig. 6b). Further deviations from the standard APOLLO installation include four loggers mounted on balconies (040UHa, 092PGa, 111OGa, and 113PGa; Fig. 6e) and two WXT stations installed on top of a container (017LMw; Fig. 6g) and on a roof-deck of the building of the Meteorological Institute (114UHw; Fig. 6h). Due to technical issues, minor changes to the instrumental set up had to be implemented during the measurement period with the replacement of one APOLLO (068OGa) and two WXT stations (048OGw and 114UHw) as well as the replacement of one APOLLO by a WXT station (083PGa).

Ensuring the implementation of FESST@HH was compatible with pandemic-related restrictions affected not only the selection of measurement sites but also the maintenance strategy. Instead of a small team servicing the entire network, the maintenance work was shared between nearly 40 people. Private landowners, who provided measurement sites in their backyard, could also volunteer to regularly change the battery, check the data, and upload it to a FTP server. Other stations located on public grounds, schoolyards or industrial premises were serviced by colleagues living nearby. The main benefit of this main-

tenance strategy was that the collective effort kept the individual workload very low and promoted the continuous operation of the instruments, which eventually eased the execution of the experiment under challenging circumstances.

## 4 Data processing

The processing of the APOLLO and WXT station measurement data is a two-stage procedure. This study describes the published level 2 data format that contains quality-controlled data in a standardized format. In contrast, level 0 data are the raw data as directly produced by the instruments and level 1 data are homogenized data for preliminary analyses, but have not passed any quality checks.

### 4.1 Level 0 data

The raw measurement data are stored in ASCII files. The APOLLO data logger creates a new file at the start time, which contains the internal logger time, the GPS-synchronized time, the resistance readings of the NTC thermometer, the pressure readings of the BME280 module and status information at 1 Hz frequency. By contrast, new level 0 WXT data files are created at midnight (UTC) on a daily basis. Each line in the files consists of a GPS-synchronized time stamp and the measurement data of a specific sensor module, whereas each module sends data at 10 s intervals.

### 4.2 Level 1 data

Measurement data, which have passed the first processing step, are called level 1 data. At this stage, basic standardization procedures are performed to allow for easy access to the data for preliminary analyses, but not quality checks. Most importantly, this includes the correction and homogenization of the data time stamps. For the APOLLO data, the internal logger time is corrected with its deviation from the most recent valid GPS time stamp that is logged once per hour. Data for missing time steps are filled with placeholder values (NaN). Furthermore, daily files are created by splitting multi-day files and merging sub-daily files. In case of the WXT data, all time stamps are moved to a regular 10 s time grid. The measurement data itself are not manipulated, except for the APOLLO thermometer readings that are converted from resistance to temperature. For both station types, level 1 data are stored in a standard ASCII format.

### 4.3 Level 2 data

The purpose of the second processing level is to generate a standardized and quality-controlled data product that is ready to use for scientific analyses. Since the raw APOLLO measurements contain different kinds of erroneous data, which originate from technical issues of the logger or characteristics of the measurement site, we apply several quality checks and corrections to clean up the data. This processing step is not required for WXT data due to internal quality checks implemented by the manufacturer of the sensor. In total, less than 1 % of the level 1 data fail the quality criteria and need to be removed. The correction procedure contains the following steps:

- removing erroneous temperature and pressure data outside plausible limits defined by $0 \leq T \leq 40\,^{\circ}\mathrm{C}$ and $950 \leq p \leq 1050\,\mathrm{hPa}$, respectively;

- removing spikes in temperature (pressure) larger than $\Delta T = 0.5\,\mathrm{K}$ ($\Delta p = 1\,\mathrm{hPa}$), where $\Delta T$ ($\Delta p$) denotes the absolute difference from the $30\,\mathrm{s}$ ($60\,\mathrm{s}$) running median value. For the temperature data, spikes are only removed for phases of active WiFi connection, which produces the spikes;

- removing temperature data showing implausibly large differences from the current network mean value defined as $|T - \overline{T}| > 15\,\mathrm{K}$. For three stations (037UHa, 040UHa, 068OGa) with erroneous NTC temperature sensors producing unphysical spikes, a criterion of $|T - \overline{T}| > 2\,\mathrm{K}$ is applied, and all data within a time window of 15 min before and after erroneous data are also removed;

- manually removing single periods of erroneous temperature and pressure measurements that are not filtered by the previous criteria. In case of station 039UHa, temperature measurements on more than 40 days are affected by the warm air outlet of a nearby air conditioning facility and are manually removed;

- applying a $10\,\mathrm{s}$ running mean smoothing procedure on the temperature measurements of station 113PGa to account for an anomalously high noise level of the NTC sensor; and

- correcting the individual biases of NTC temperature sensors. The biases are determined from one week long calibrations with respect to a reference WXT sensor (114UHw) between January and June 2020. The pressure readings of the BME280 sensors are not calibrated since in test measurements the absolute values showed an oscillating drift pattern with an amplitude and period of about $0.5\,\mathrm{hPa}$ and 10 days. The WXT536 and Pt1000 sensors are calibrated by the manufacturer.

The level 2 data (Kirsch et al., 2021b) are stored in the NetCDF4 data format (Eaton et al., 2017), while the naming of files, variables and attributes complies with the SAMD (Standardized Atmospheric Measurement Data) Product Standard (Lammert et al., 2018). Accordingly, all measurement data of the same variable and from the same type of instrument are merged and stored in daily files. Note that the time variable expressed as seconds since 1 January 1970, 00:00:00 UTC ignores leap seconds (i.e., occasional adjustments of UTC time to variations in Earth's rotation velocity). Also note that the time drift correction and quality control procedures cause irregular and isolated data gaps that may affect only single seconds which has to be taken into account when using the data set at its full time resolution. The file header contains all relevant metadata describing the individual measurement sites, namely the station identifier, the station name, the geographic coordinates, the altitude above sea level, the sensor height above ground, and the LCZ. Attached to the data set are also images of all measurement stations and panoramic views of the surroundings of all WXT stations to support the interpretation of the corresponding observation data.

## 5  Description of data set

### 5.1  Data availability

The level 2 data set of FESST@HH covers the period from 1 June to 31 August 2020 (Fig. 7a). The average availability of valid temperature observations is 83.2 % and 87.6 % for all APOLLO and WXT stations, respectively. Since the 3 month period also includes the installation phase of the instruments during the first half of June, these numbers increase to 90.0 % and 94.3 % if only the period after 15 June is considered. During this period, the daily availability of APOLLO and WXT measurements is always larger than 82.6 % and 90.0 %, respectively. Apart from the removed measurements affected by erroneous sensors and local site conditions, most of the data gaps in the APOLLO data are due to missing power supply of the loggers, either caused by technical issues related to the power bank batteries or insufficient maintenance. In only one case was a battery stolen. The greater stability of the WXT power supply is also the reason for the generally greater availability of WXT data compared to APOLLO data.

### 5.2  Weather conditions

The weather conditions during the FESST@HH period covered the full range of a typical mid-latitude summer. The air temperature measured by the station network ranges between a minimum of 5.3 °C on 13 July and a maximum of 35.6 °C on 8 August (Fig. 7b; days defined in UTC). During the entire 92 day period, the maximum temperature exceeded 25 °C on 41 days and 30 °C on 16 days, including an exceptionally long period of 12 consecutive days with maximum daily temperature exceeding 30 °C between 6 August and 17 August. In contrast, July was characterized by relatively cold temperatures, with a maximum of above 25 °C on only 6 days. Measurable rainfall, defined as a daily rainfall accumulation of more than 1 mm at at least one WXT station of the network, was observed on approximately half of the days from 1 June to 31 August (Fig. 7c). On 11 days, mostly between 27 June and 10 July, the median rainfall amount exceeded 5 mm. A maximum daily accumulation of 28.2 mm was observed on 27 June in association with a strong convective event.

## 6  Observations of sub-mesoscale phenomena

The goal of the FESST@HH experiment was to design and operate an observation network that is dense enough to investigate the structure of convective cold pools. However, the data set also contains information and potential scientific implications for additional sub-mesoscale phenomena that we present here.

### 6.1  Cold pools

During the 3 month measurement period, the station network recorded 37 cold pool events. We define events as periods with at least five stations per hour satisfying the cold pool detection criterion described in Kirsch et al. (2021a), i.e., a local temperature drop of at least 2 K within 20 min. This rather conservative threshold was chosen to robustly discriminate the cold pool signals from other potential sources of spatial temperature variability. The plausibility of cold pool detections is checked based on the

visual inspection of corresponding rain radar imagery by a X-band radar of the University of Hamburg (Burgemeister et al., 2022) and the operational C-band radar network of the German Weather Service. The number of approximately 12 events per month is larger than the long-term average at the Hamburg weather mast site of about 7 events per month (Kirsch et al., 2021a) as expected from the larger areal coverage compared to a point measurement. Fig. 8 shows an overview of all 37 identified cold pool events and illustrates the spatial variability of maximum temperature perturbations detected within the station network. Most of the events (28) exhibit a median temperature signal of -4 K or weaker, however, the variability in signal strength and number of affected stations largely vary for different events. To demonstrate the ability of the station network to capture the characteristics of a individual cold pool and to shed light on its internal structure, we present one case study.

On 10 August 2020 around 12:45 UTC, a strong and nearly stationary convective precipitation cell developed southeast of the city center. About 10 min later, the formation of a cold pool became evident from a rapid cooling of the surface-layer air. Fig. 9 illustrates the evolution of the cold pool with snapshots of the temperature perturbations relative to the pre-event state observed by the station network. About 20 min after initiation of the convective cell, the cold pool exhibited a temperature perturbation of up to -8 K within an area of less than 10 km in diameter (Fig. 9a). After another 20 min, the temperature perturbation strengthened to about -11 K and the cold pool expanded to a size of roughly 10 km×20 km (Fig. 9b). This process continued until around 14:00 UTC when the cold pool reached its maximum temperature perturbation of about -12 K and a diameter of nearly 30 km when assuming a roughly elliptical shape of the cold pool, whereas its northeastern parts were outside of the network (Fig. 9d). The associated near-surface wind field observed by the WXT weather stations indicated a strong divergent flow at the cold pool center, especially during the early stages of the cold pool's life cycle (Fig. 9a and b). Consistent with expected characteristics of a cold pool, the radial expansion of the cold-air region during the later stages was also present in the wind observations, predominantly southwest of the cold pool center (Fig. 9c and d).

The time series of temperature measurements at selected stations of the network provide further insights into the spatial structure and life cycle of the cold pool (Fig. 10a). APOLLO station *Luxweg* (104PGa) near the cold pool center experienced an initial temperature drop of approximately 8 K within 5 min that continued at a slower rate afterwards and reached a maximum perturbation of -12 K after one hour. This value is on the order of strongest temperature perturbations expected during cold pool passages in Hamburg (Kirsch et al., 2021a). The readings of the stations *Ochsenwerder Norderdeich* (082PGa) and *Obsthof Lehmbeck* (054OGa), located 4 km and 12 km further south, respectively, reveal that the maximum amount of cooling decreases by about two-thirds away from the center, which suggests a highly heterogeneous temperature structure inside the cold pool. Since the site *Luftmessnetz Habichtstraße* (018LMa), situated northwest of the cold pool center, experienced no cooling at all, the cold pool produced a gradient in temperature perturbation of 12 K within a distance of 11 km. This result is in line with the findings of van den Heever et al. (2021), who report a variation of near-surface cold pool temperature perturbations of order 1 to 2 K on scales between $\mathcal{O}(100)$ m and $\mathcal{O}(1)$ km. The time lags between the cooling signals observed on the southern side of the cold pool also allow us to estimate its propagation velocity. Based on time lags of about 15 min and 50 min for travel distances of 3.9 km (104PGa–082PGa) and 11.9 km (104PGa–054PGa), respectively, the velocity is about 4 m s$^{-1}$. This value is lower than the propagation velocity of 6.7 m s$^{-1}$ found for a cold pool event in north-central Oklahoma (Borque et al., 2020),

but plausible considering that the parent precipitation cell was almost stationary and, therefore, the observed propagation of the cold outflow air was not biased by the movement of the cell itself.

The accompanying air pressure measurements at the four selected sites confirm that the negative temperature perturbation of the cold pool is associated with a typical hydrostatic pressure rise (Fig. 10b). However, the positive pressure perturbation was well-pronounced only at station 104PGa, while the amplitudes were not proportional to the strength of the corresponding cooling signals. The pressure signal at station 104PGa also experienced rapid fluctuations of about $1\,\mathrm{hPa}$ that were not apparent at the three other sites. Although an in-depth investigation of such effects lies beyond the scope of this study, this may indicate that the spatial structure of a cold pool differs between temperature and pressure and that mechanisms other than hydrostatic cooling control the pressure signal near the center, like dynamic effects already described in early studies (i.e., Wakimoto, 1982; Mueller and Carbone, 1987).

## 6.2 Urban heat island

The FESST@HH data set is well suited for studying urban climate effects, as the station network covers both the city center and its rural surroundings. Stations near the city center record weather conditions with urban modifications, whereas rural stations provide undisturbed references. Focusing on air temperature, we define the urban modification $\Delta T_{\mathrm{city}}$ as the difference between the locally observed air temperature and the undisturbed, natural temperature $T_{\mathrm{nat}}$, determined exclusively by large-scale weather conditions. On scales of $\mathcal{O}(10)\,\mathrm{km}$ without orography, it is reasonable to assume that $T_{\mathrm{nat}}$ varies only linearly in space. We estimate the mean and the slope of this linear temperature field with a regression using all stations in weakly sealed (i.e., mostly natural) environments, defined by a maximum sealing up to *open low-rise* (LCZ 6). The mean diurnal cycle of $\Delta T_{\mathrm{city}}$ features a step function at almost all stations (Fig. 11a). During daytime, the urban effects are small and $\Delta T_{\mathrm{city}}$ is close to zero everywhere. In contrast, during nighttime, predominantly between 21 and 3 UTC, $\Delta T_{\mathrm{city}}$ reaches a constant level at all stations, which largely reflects the well-known nocturnal urban heat island effect. For example, the most central station *Wetterstation HafenCity* (010WSa; orange in Fig. 11a) is almost 2.5 K warmer than estimated $T_{\mathrm{nat}}$ and the most rural site *Obsthof Schuback* (055OGw) is even 1.5 K colder than the reference. The spatial pattern of the mean nocturnal urban heat excess is in agreement with the typical heat island structure featuring largest values close to the city center (Fig. 11b). Further analysis of individual days (not shown) provides indication that the advection of the heat island effect causes downwind heat plumes to form, which affect specific outskirts depending on the prevailing wind direction.

## 6.3 Turbulent temperature fluctuations

The NTC temperature sensor of the APOLLO station responds rapidly to air temperatures fluctuations. The recorded time series with a sampling rate of $1\,\mathrm{Hz}$ partly resolves temperature eddies related to turbulent sensible heat fluxes. The high frequency temperature fluctuation expressed by the standard deviation of temperature readings within $1\,\mathrm{min}$ intervals, $\sigma_{\mathrm{T}}$, features a well-pronounced diurnal cycle (Fig. 12a). During nighttime, $\sigma_{\mathrm{T}}$ is close to 0.05 K at all stations, which most likely resembles the noise level of the instruments. However, at daytime, $\sigma_{\mathrm{T}}$ is well above that noise level and follows the common diurnal cycle of net radiation or sensible heat flux at the surface. Turbulent fluxes are strongly determined by local surface

conditions like albedo, sealing or vegetation cover. Likewise, the differences in the diurnal cycle of $\sigma_T$ can be explained by local surface conditions: the station *Stadtteilschule Blankenese* (033KBa) at a schoolyard next to a sportsground (low albedo, no evapotranspiration) with free insolation exhibits the highest temperature variability. High values also occur at the schoolyard of station *Schule Redder* (036KBa) but are shifted and limited to morning hours due to a building west of the station casting a shadow and reflecting sunlight. Even more extreme, buildings shade the station *Luftmessnetz Rothenburgsort* (020LMa) almost the whole day except during the late afternoon. The lowest temperature variance is observed at station *Uni Sternwarte* (043UHa), the only station below tall trees. The heat exchange within the tree canopy effectively reduces the energy exchange at the ground.

The relation between surface heat exchange and temperature variability is also directly confirmed by correlating the net radiation observed at the Hamburg weather mast reference site (Bruemmer et al., 2012) with $\sigma_T$ at the neighboring APOLLO station *Wetterstation Zollamt* (011WSa) located about 750 m away (Fig. 12b). Values of $\sigma_T$ lower than 0.75 K are strongly correlated to situations where the net radiation is close to zero. Increasing variance beyond this threshold is mostly explained by increasing net radiation ($r^2$=0.77). Instead of the sensible heat flux, we use the net radiation for this analysis to minimize the influence of land surface differences between the two sites (pasture at *Wetterstation Zollamt* versus sealed area at Hamburg weather mast). Additional measurements on a test site show that $\sigma_T$ does not change if the sensor shield is protected from direct solar radiation (not shown). This finding supports our reasoning that the APOLLO stations capture turbulent temperature fluctuations caused by the surface heat exchange in the near environment of the observation site rather than artificially generated by the heating of the sensor shield. Direct measurements of turbulent heat fluxes and surface properties like temperature and moisture added in the FESSTVaL follow-up experiment will help to better understand the land–atmosphere interactions for different land classes.

# 7   Summary and conclusions

The FESST@HH field experiment took place in Hamburg (Germany) from June to August 2020 with the primary aim of illuminating the internal structure of convective cold pools that conventional observations miss. To this end, a dense network of 103 autonomously operating weather stations was installed over an area of $50\,\mathrm{km}\times35\,\mathrm{km}$ with an average distance between the stations of 1.85km. The measurements were mainly conducted by 82 low-cost, custom-built APOLLO stations that were designed to sample the strong and rapid changes in air temperature and pressure associated with cold pools at 1 s resolution. Additional measurements of relative humidity, wind speed and precipitation at 10 s resolution were performed by 21 WXT weather stations based on commercial sensors.

The FESST@HH data set is unique not only with respect to its scientific potential for investigating sub-mesoscale atmospheric processes but also with respect to its implementation. The experiment was successfully conducted despite the exceptional circumstances caused by the COVID-19 pandemic, especially thanks to the short-notice support of local institutions and private landowners, who provided their grounds as measurement sites and helped with the station maintenance. The low-cost and self-manufactured measurement instruments operated smoothly without any major technical problems, although they had

never been used in such a large number before. The good performance is reflected by an availability of quality-controlled data of more than 90 % during the main observation period. Most importantly, a case study demonstrates that the network is capable of mapping the sharp horizontal temperature gradients produced by a convective cold pool and capturing its spatial footprint. Throughout its life cycle, the cold pool diameter grows from less than $10\,\mathrm{km}$ to nearly $30\,\mathrm{km}$, while its leading edge propagates at a velocity of about $4\,\mathrm{m\,s^{-1}}$ away from the center of convection. There is also evidence that the spatial variance of the corresponding pressure signals differs from the cooling signature and may indicate the presence of non-hydrostatic effects acting within the cold pool center. Furthermore, the data set is relevant for studies in urban meteorology, as the dense temperature observations include information on the spatial structure of the nocturnal urban heat island as well as on the local surface heat exchange mirrored by high-frequency temperature fluctuations.

The FESST@HH field experiment proves that it is possible to conduct observations that close the information gap left by conventional measurement networks. In addition, it highlights the real need for dense station networks that act like a magnifying glass for revealing sub-mesoscale atmospheric processes. The use of a large number of low-cost instruments designed for a specific purpose is a feasible strategy to tailor a measurement network that is dense enough to shed light on previously unobserved scales. These data are scientifically valuable not only for a deeper understanding of cold pools and the convective life cycle, but also for the validation of hectometer-scale numerical simulations. Moreover, the data set leaves space for unexpected results and applications not yet anticipated. FESST@HH also exemplifies how a major community effort can ease the execution of a major scientific enterprise or, as in this case, make it possible at all. In this sense, FESST@HH has the potential to be a prototype for future field campaigns. It already provides a proof of concept for an extended investigation of cold pools and further sub-mesoscale boundary layer structures during the FESSTVaL 2021 experiment.

## 8  Data availability

The data set is available for download at https://doi.org/10.25592/UHHFDM.10172 (Kirsch et al., 2021b). The data are stored in netCDF files per station type, measurement variable and day. The file size is about $25\,\mathrm{MB}$ and $1.5\,\mathrm{MB}$ for APOLLO and WXT data, respectively. For both station types all daily files are merged into monthly *.zip files. The data set also includes a file containing images of all measurement stations and panoramic views of the surroundings of all WXT stations.

*Author contributions.* BK: methodology (design, assembling and testing of instruments), investigation (installation and operation of instruments), data curation (compilation of meta data and processing of data), formal analysis (except for sections 6.2 and 6.3), writing - original draft (except for sections 6.2 and 6.3), writing - review and editing; CH and DK: conceptualization, supervision, project administration; RS: methodology (design, assembling and testing of APOLLO stations), software (software development for APOLLO stations); MO: methodology (design, assembling and testing of instruments), investigation (installation and operation of measurement stations); FA: conceptualization, supervision, methodology (design of instruments), investigation (installation and operation of measurement stations), formal analysis (sections 6.2 and 6.3), writing - original draft (sections 6.2 and 6.3)

*Competing interests.* The authors declare that they have no conflicts of interests.

*Acknowledgements.* The authors are grateful to Marco Clemens and Ingo Lange for heavily supporting the manufacturing, installation and maintenance of measurement stations and to Sarah Wiesner for planning and coordinating the field experiment. We thank Theresa Szyszka and Jan Moritz Witt for assembling the APOLLO logger main boards, manufacturing the components of the APOLLO and WXT stations and supporting their maintenance. For either helping with the installation and maintenance of stations or providing ground for measurement sites we are thankful to Sigrid and Werner Bock, Agnes Bornholdt, Frank Böttcher, Burghardt Brümmer, Stefan Bühler, Finn Burgemeister, Guido

Cioni, Martin Claussen, Traute Crüger, Henning Dorff, Tobias Finn, Veronika Gayler, David Grawe, Akio Hansen, Anja Hermans, Tatiana Ilyina, Olaf Jäke, Markus Kilian, Stefan Kinne, Marlene and Reinhard Kirsch, Brita Kliemt, Hartmut Kock, Heike Konow, Sascha Krueck, Heiner Meier, Ulrike Niemeier and Luis Kornblueh, Peter Peitzner, Jon Petersen, Thomas Raddatz, Jule Radtke, Berinike Rappat, Florian Römer, Nicole and Steffen Rüeß, Rita Schierholz, Imke Schirmacher, Hauke Schmidt, Amelie Schmitt, Hauke Schulz, Constanze Seibert, Milica Stankovic, Claudia Timmreck and Hannes Thiemann, Karla Tomhawe, Bodo Voigt, Stefan Walter, Ralf Wedemann, Tobias Weiß, Su-

sanne and Heinz-Dieter Wenck, Benjamin Will, Holger Witt and Florian Ziemen. We acknowledge the following institutions, schools, clubs and companies for allowing us to install measurement stations on their grounds: Hamburger Luftmessnetz, Universität Hamburg, Bundesanstalt für Wasserbau Hamburg, Deutscher Wetterdienst Niederlassung Hamburg, Drachenfreunde Winsen, Greenpeace Wilhelmsburg, Gut Karlshöhe, Gymnasium Rahlstedt, Halepaghen-Schule Buxtehude, Hamburger Aero-Club Boberg, Hamburger Bogenschützen Gilde, Hansa Sportverein Stöckte, Hochschule für Angewandte Wissenschaften Hamburg, Immanuel-Kant-Gymnasium Harburg, Obsthof Axel Schuback,

Obsthof Lehmbeck, Overmeyer Landbaukultur, Restaurant Bobby Reich, Restaurant Fährhaus Tatenberg, Schule Redder Sasel, Segelclub Rhe Hamburg, Segelvereinigung Sinstorf, Stadtteilschule Blankenese, Stadtteilschule Eidelstedt, Stadtteilschule Meiendorf, Suntrace, Tennisclub Eichtalpark and TSV Holm. We also thank all persons and institutions that supported the experiment in any way and were not explicitly mentioned here.

This research was carried out in the Hans Ertel Center for Weather Research (Hans-Ertel-Zentrum für Wetterforschung; HErZ). This
German research network of universities, research institutions, and the German Weather Service (Deutscher Wetterdienst; DWD) is funded by the Federal Ministry of Transport and Digital Infrastructure (Bundesministerium für Verkehr und digitale Infrastruktur; BMVI).

BK is thankful to Jochem Marotzke and Dallas Murphy for providing valuable tips on the writing style and structure of the manuscript and to Arjun Kumar for proofreading.

The authors thank Aryeh Jacob Drager and one anonymous referee for carefully reviewing the manuscript and for their detailed and
435 constructive comments.

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

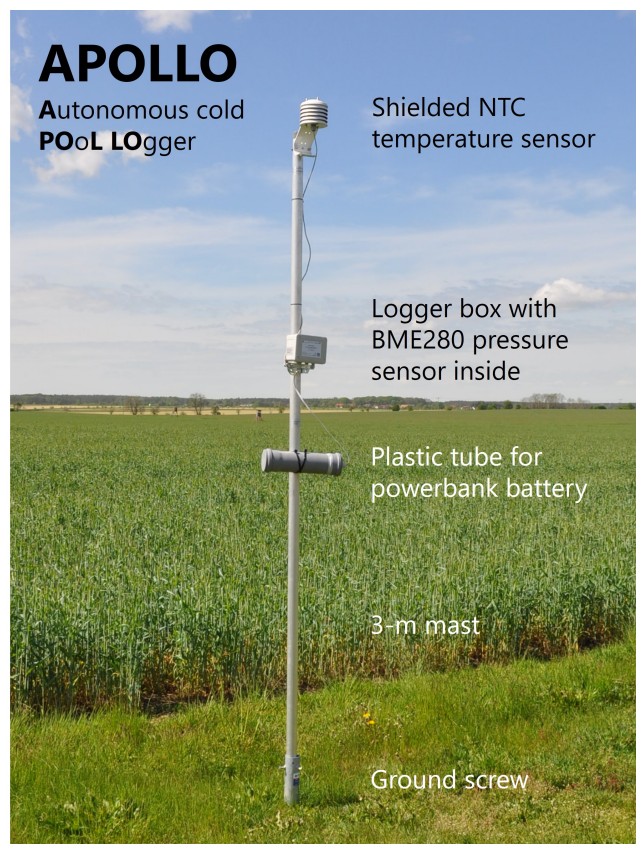

**Figure 1.** Components of APOLLO station.

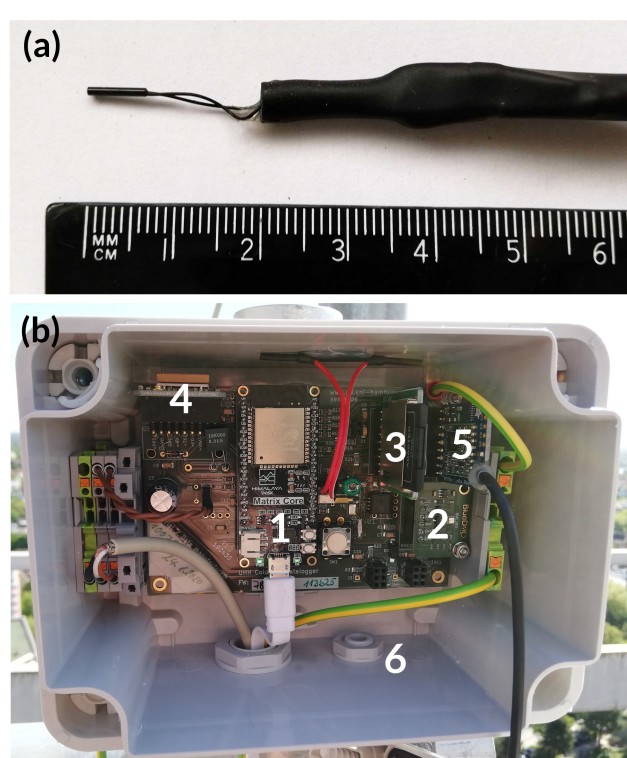

**Figure 2.** (a) NTC thermometer with scale for reference, (b) logger main board of APOLLO station. Marked are the micro controller board (1), digital air pressure sensor BME280 (2), micro-SD memory card (3), GPS receiver module (4), LoRa module (5) and pressure counterbalance valve (6).

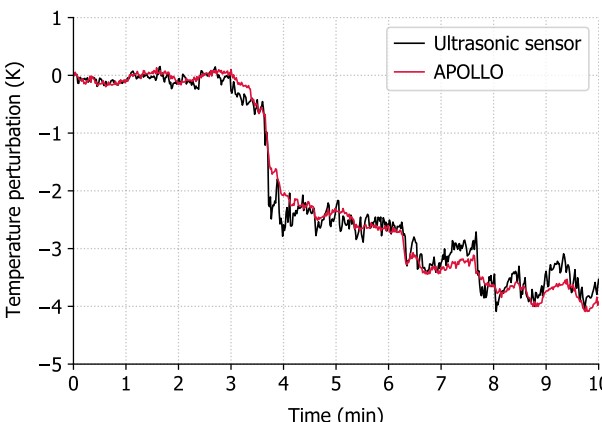

**Figure 3.** Air temperature perturbation observed by a co-located ultrasonic sensor and APOLLO station during a cold pool event at the Meteorological Observatory Lindenberg (eastern Germany) on 26 August 2019. The readings of the ultrasonic sensor are 1 s averages of 20 Hz acoustic temperature measurements by an ultrasonic anemometer (type: METEK uSonic-3 Scientific).

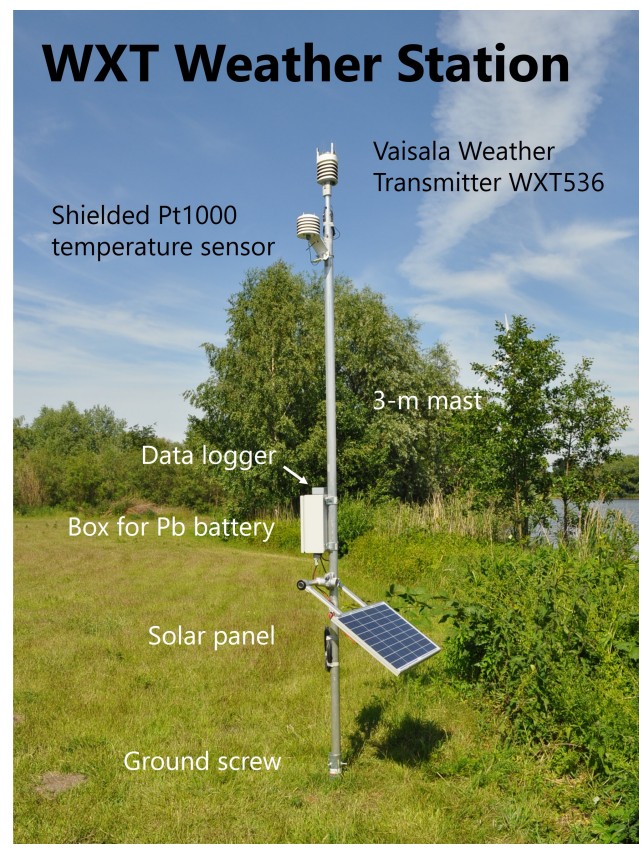

**Figure 4.** Components of WXT weather station.

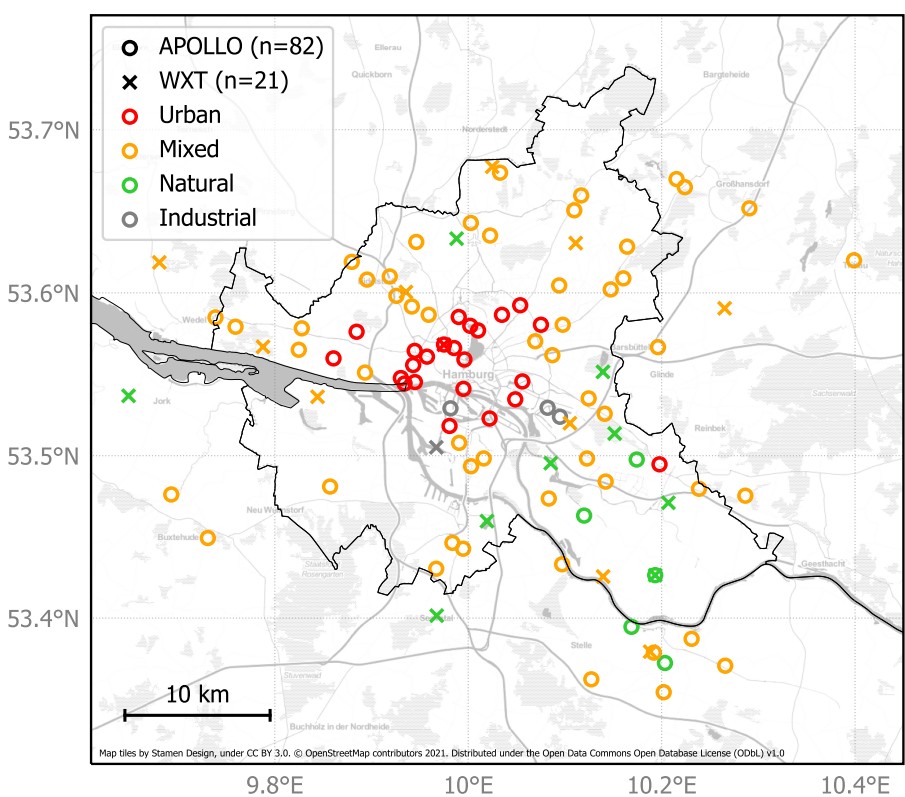

**Figure 5.** Map of measurement locations of APOLLO and WXT weather stations during the FESST@HH 2020 experiment. Colors indicate environmental conditions of measurement sites grouped into urban (local climate Zone 2, 4 or 5), mixed (LCZ 6), natural (LCZ 9, B or D) and industrial (LCZ 8 or 10) following the framework of Stewart and Oke (2012). For stations falling into two LCZs, only the predominant sub-class is depicted. Black lines mark the city limits of Hamburg.

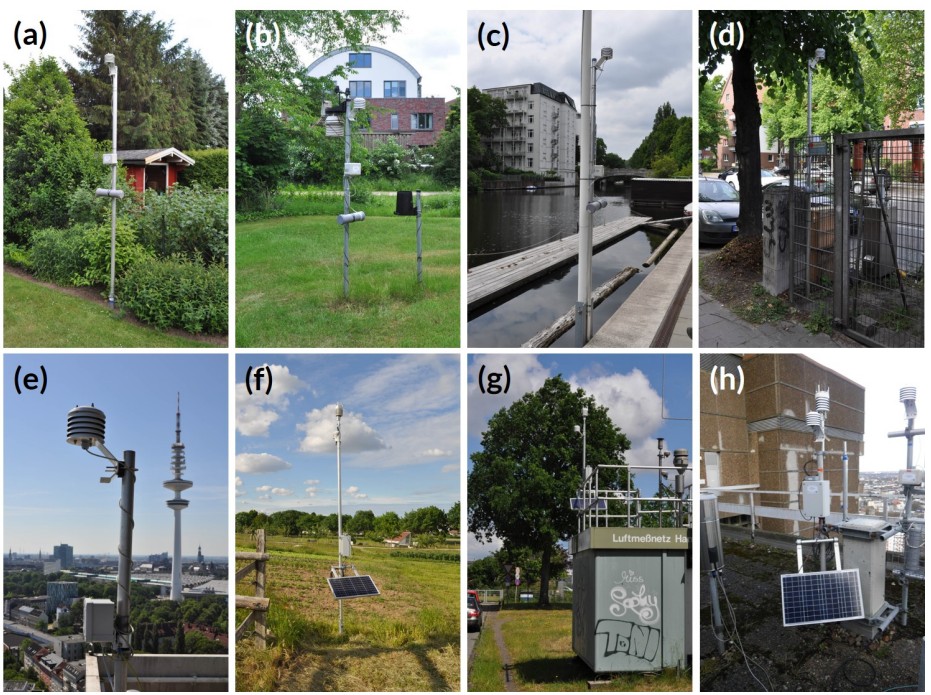

**Figure 6.** Examples of instrument installations during FESST@HH: APOLLO station (a) as the standard installation in a private garden, (b) attached to an existing weather station on a schoolyard, (c) on a boat landing stage at a channel, (d) on an air quality observation site at a public road, (e) on the balcony of the building of the Meteorological Institute, and WXT weather station (f) as the standard installation on an agricultural field, (g) on top of a container, and (h) on the roof-deck of the building of the Meteorological Institute.

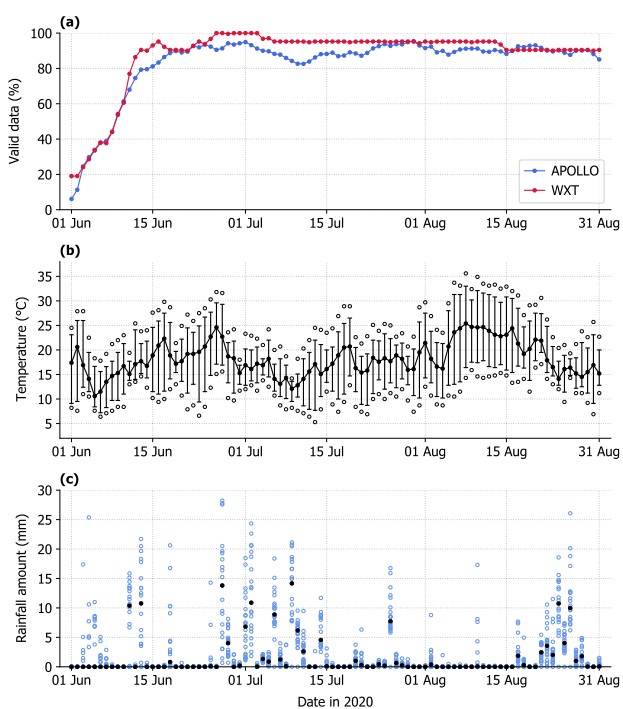

**Figure 7.** FESST@HH level 2 data from 1 June to 31 August 2020: (a) Daily mean fraction of valid temperature observations over all APOLLO and WXT stations, (b) daily median (filled dots), 5th percentile and 95th percentile (whiskers) and minimum and maximum (open circles) of all WXT air temperature observations, (c) individual accumulated rainfall amount at all WXT stations (open circles). Filled dots mark the daily median. Days are defined in UTC.

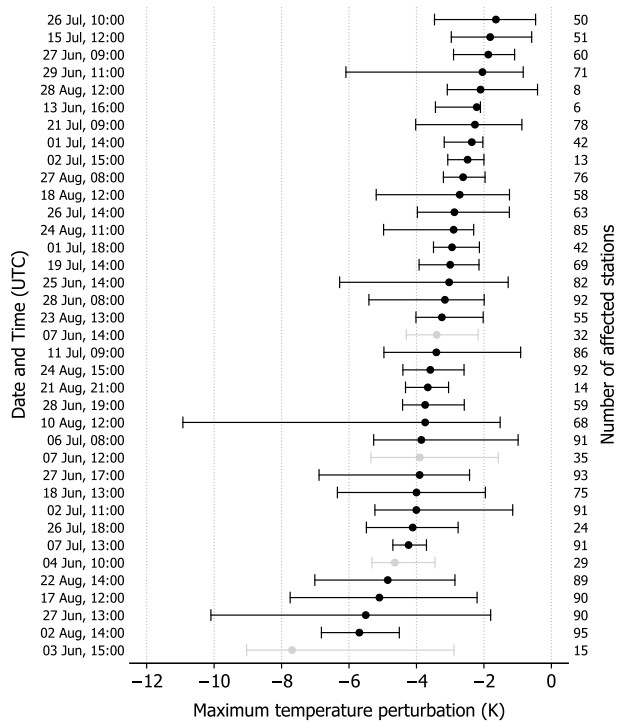

**Figure 8.** Distribution of the maximum temperature perturbation detected at individual APOLLO and WXT stations for the 37 identified cold pool events during FESST@HH. Blacks dots and whiskers mark the median value and 5th/95th percentiles, respectively. The events are sorted according to the median values in maximum temperature perturbation. Events with an average data availability over all stations of less than 75 % are marked in grey. The times specify the defined begin of the respective events about 30 to 60 min prior to the onset of rainfall, whereas the duration of an event is 4 hours. Indicated is also the number of stations that have experienced a cold pool passage.

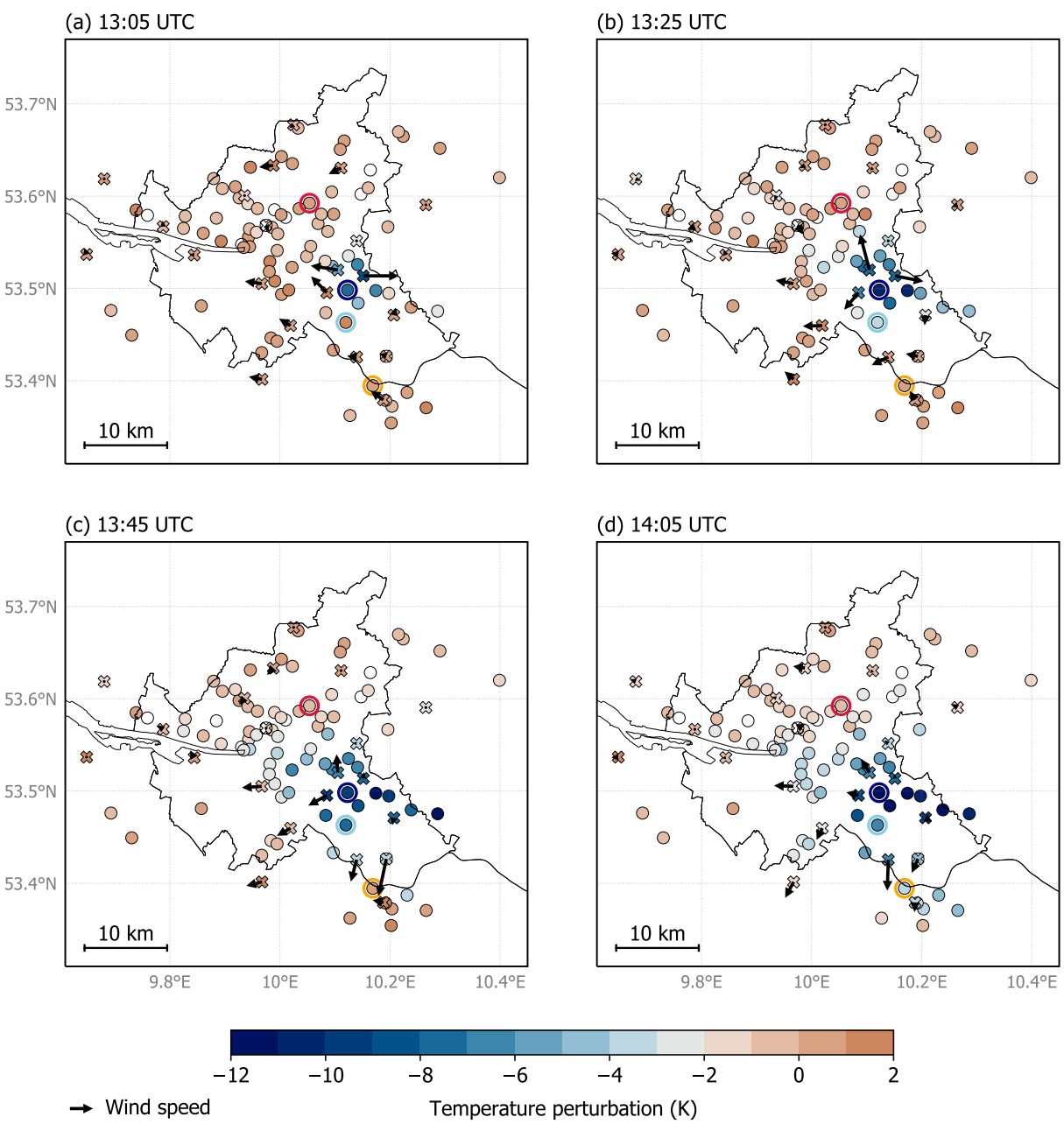

**Figure 9.** Perturbation in air temperature since 12:30 UTC from (a) 13:05 UTC to (d) 14:05 UTC observed by APOLLO (circles) and WXT stations (crosses) during a cold pool event on 10 August 2020. Bluish colors mark stations inside the cold pool, defined by a temperature perturbation stronger than -2 K (color map by Crameri (2018); Crameri et al. (2020)). Black arrows indicate instantaneous wind speed and direction observed by WXT stations. The length of the reference arrow corresponds to $5\,\mathrm{m\,s^{-1}}$ wind speed. The four highlighted stations refer to the respective data illustrated in Fig. 10.

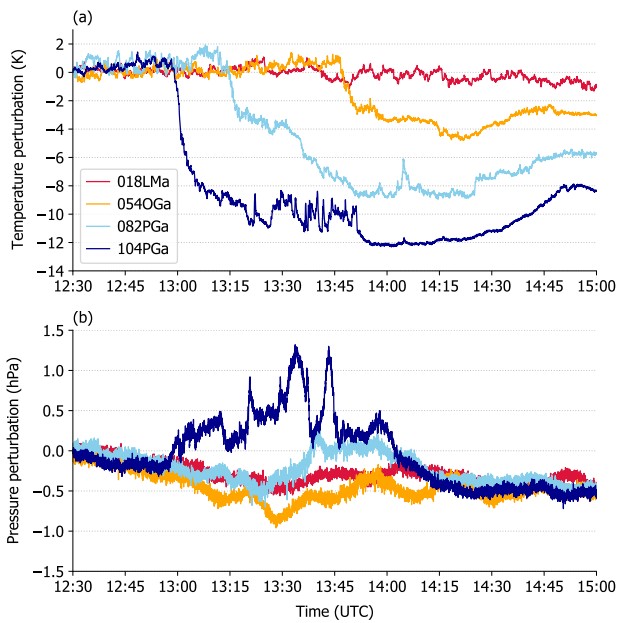

**Figure 10.** Perturbation in (a) air temperature and (b) pressure since 12:30 UTC observed by the four selected APOLLO stations *Luftmessnetz Habichtstraße* (018LMa), *Obsthof Lehmbeck* (054OGa), *Ochsenwerder Norderdeich* (082PGa) and *Luxweg* (104PGa) during a cold pool event on 10 August 2020. Locations of the stations are highlighted in Fig. 9.

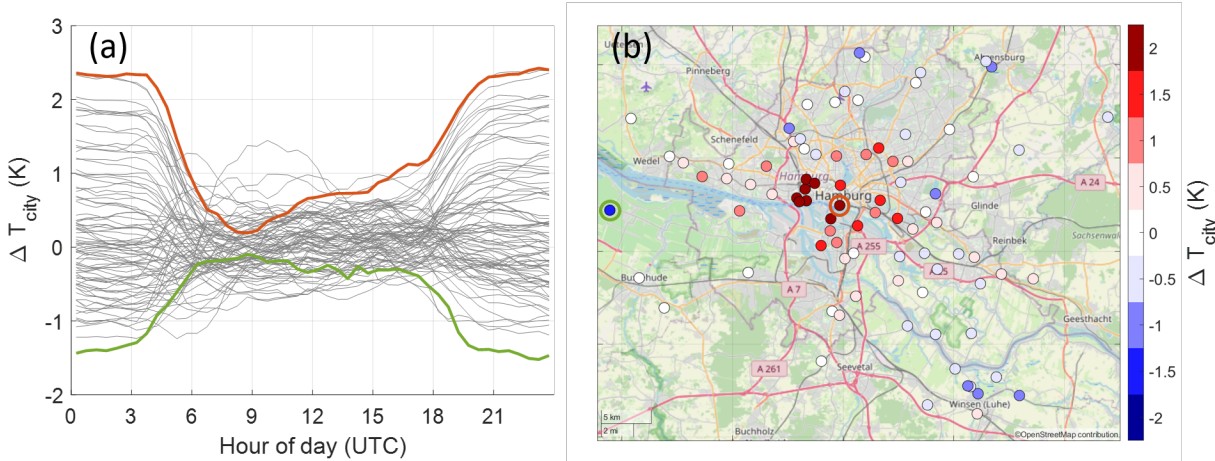

**Figure 11.** (a) Mean diurnal cycle of urban temperature modification $\Delta T_{\mathrm{city}}$ and (b) nighttime (21:00 UTC to 03:00 UTC) mean of $\Delta T_{\mathrm{city}}$ at all APOLLO and WXT stations from 15 June to 31 August 2020. Highlighted in orange and green are the urban station *Wetterstation HafenCity* (010WSa) and the rural station *Obsthof Schuback* (055OGw), respectively. © OpenStreetMap contributors 2021. Distributed under the Open Data Commons Open Database License (ODbL) v1.0.

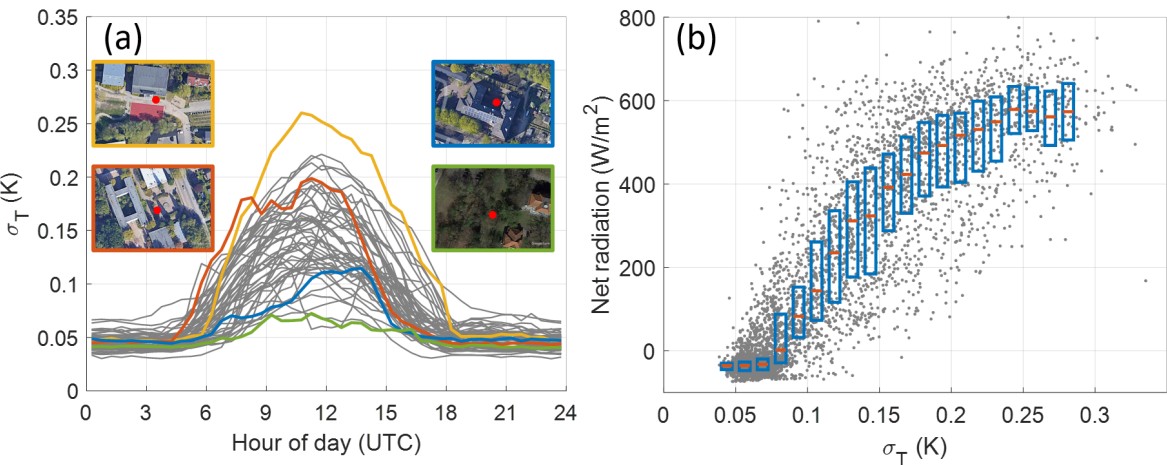

**Figure 12.** (a) Mean diurnal cycle of standard deviation of $1\,\mathrm{min}$ temperature fluctuations $\sigma_T$ at all APOLLO stations (grey lines) and at the highlighted stations *Stadtteilschule Blankenese* (yellow; 033KBa), *Schule Redder* (orange; 036KBa), *Luftmessnetz Rothenburgsort* (blue; 020LMa) and *Uni Sternwarte* (green; 043UHa). Map data © Google Earth 2021. (b) Net radiation at the Hamburg weather mast site dependent on $\sigma_T$ at station *Wetterstation Zollamt*, both averaged over $10\,\mathrm{min}$. Blue boxes and orange bars indicate the inter-quartile range and median for the respective $\sigma_T$ bin. The analyses in both (a) and (b) are valid for sunny days from 1 June to 31 August 2020, whereas sunny days are defined as days on which the daily averaged global radiation at Hamburg weather mast is larger than at least one-third of the daily averaged theoretical maximum of clear sky insolation.

**Table 1.** Sensor specifications of APOLLO station (TE Connectivity, 2015; Bosch, 2020).

| Parameter | Sensor | Measurement range | Accuracy |
|---|---|---|---|
| Temperature | NTC thermistor | -40 ... 100 °C | ±0.2 K at 0 ... 70 °C |
| Pressure | BME280 digital sensor | 300 ... 1100 hPa | ±1 hPa at 0 ... 30 °C (absolute) |
| | | | ±0.12 hPa at 25 ... 40 °C (relative) |

**Table 2.** Sensor specifications of WXT weather station (Vaisala, 2020).

| Parameter | Sensor | Measurement range | Accuracy |
|---|---|---|---|
| Temperature | PTU module | -52 ... 60 °C | ±0.3 K at 20 °C |
| | Pt1000 thermometer | -50 ... 85 °C | ±1.3 K at 20 °C |
| Pressure | PTU module | 500 ... 1100 hPa | ±0.5 hPa at 0 ... 30 °C |
| | | | ±1.0 hPa at -52 ... 60 °C |
| Relative humidity | PTU module | 0 ... 100 %RH | ±3 %RH at 0 ... 90 %RH |
| | | | ±5 %RH at 90 ... 100 %RH |
| Wind speed | Ultrasonic anemometer | 0 ... 60 m s$^{-1}$ | ±3 % at 10 m s$^{-1}$ |
| Wind direction | Ultrasonic anemometer | 0 ... 360° | ±3° at 10 m s$^{-1}$ |
| Rainfall | Piezoelectrical sensor | - | better than 5 %, weather dependent |

**Table 3.** Information about APOLLO measurement sites. Altitude denotes the height of ground above sea level and sensor height is the height of temperature sensor above ground. See section 3.1 for explanation of local climate zones (LCZ) after Stewart and Oke (2012).

| Identifier | Name | Latitude (°N) | Longitude (°E) | Altitude (m) | Sensor height (m) | LCZ |
|---|---|---|---|---|---|---|
| 007WSa | Wetterstation Stellingen Wohngebiet | 53.59817 | 9.92575 | 21 | 3 | 6 |
| 008WSa | Wetterstation Wilhelmsburg Grünfläche | 53.49349 | 10.00290 | 3 | 3 | 6 |
| 010WSa | Wetterstation HafenCity | 53.54109 | 9.99505 | 8 | 2 | 5 |
| 011WSa | Wetterstation Zollamt | 53.52393 | 10.09486 | 6 | 3 | 8 |
| 015LMa | Luftmessnetz Max-Brauer-Allee | 53.55571 | 9.94306 | 25 | 3 | 2 |
| 016LMa | Luftmessnetz Kieler Straße | 53.56440 | 9.94464 | 17 | 3 | 2 |
| 018LMa | Luftmessnetz Habichtstraße | 53.59240 | 10.05374 | 13 | 3 | 5 |
| 019LMa | Luftmessnetz Elbhang | 53.54523 | 9.94488 | 23 | 3 | 5 |
| 020LMa | Luftmessnetz Rothenburgsort | 53.53463 | 10.04861 | 4 | 3 | 5 |
| 021LMa | Luftmessnetz Stresemannstraße | 53.56086 | 9.95733 | 21 | 3 | 2 |
| 022LMa | Luftmessnetz Neugraben | 53.48101 | 9.85715 | 3 | 3 | 6 |
| 023LMa | Luftmessnetz Wilhelmsburg | 53.50789 | 9.99059 | 3 | 3 | 6 |
| 024LMa | Luftmessnetz Hafen | 53.52910 | 9.98163 | 6 | 3 | 8 |
| 025LMa | Luftmessnetz Veddel | 53.52287 | 10.02204 | 5 | 3 | 5G |
| 026LMa | Luftmessnetz Billbrook | 53.52945 | 10.08207 | 5 | 3 | 8G |
| 027KBa | Saselbekstraße, Sasel | 53.65976 | 10.11679 | 30 | 3 | 6 |
| 028KBa | Halepaghen Schule, Buxtehude | 53.47613 | 9.69293 | 3 | 2 | 6 |
| 029KBa | Stadtteilschule Eidelstedt | 53.60816 | 9.89548 | 19 | 3 | 6 |
| 030KBa | Stadtteilschule Meiendorf | 53.62835 | 10.16431 | 40 | 2 | 6 |
| 031KBa | Gymnasium Rahlstedt | 53.60212 | 10.14735 | 31 | 2 | 6 |
| 033KBa | Stadtteilschule Blankenese | 53.56509 | 9.82479 | 39 | 3 | 6 |
| 036KBa | Schule Redder, Sasel | 53.65064 | 10.10965 | 38 | 2 | 6 |
| 037UHa | Uni Physik DESY, Bahrenfeld | 53.57615 | 9.88457 | 38 | 3 | 5 |
| 039UHa | Uni Campus, Rotherbaum | 53.56615 | 9.98540 | 12 | 3 | 5 |
| 040UHa | Geomatikum, Balkon 16. Stock | 53.56831 | 9.97495 | 16 | 68 | 4 |
| 041UHa | Uni Botanischer Garten, Klein Flottbek | 53.55979 | 9.86068 | 29 | 3 | 5 |
| 043UHa | Uni Sternwarte, Bergedorf | 53.47964 | 10.23859 | 31 | 3 | 6A |
| 046OGa | BAW Rissen | 53.58492 | 9.73867 | 11 | 3 | 6A |
| 049OGa | Horner Rennbahn | 53.56176 | 10.08700 | 14 | 3 | 6 |
| 051OGa | Drachenwiese Winsen | 53.37231 | 10.20341 | 4 | 3 | B |
| 052OGa | HAW Campus Bergedorf | 53.49469 | 10.19780 | 19 | 3 | 5 |
| 053OGa | Hamburger Bogengilde, Fuhlsbüttel | 53.64289 | 10.00270 | 21 | 3 | 6D |
| 054OGa | Obsthof Lehmbeck, Hoopte | 53.39460 | 10.16881 | 3 | 3 | 9 |

| Identifier | Name | Latitude (°N) | Longitude (°E) | Altitude (m) | Sensor height (m) | LCZ |
|---|---|---|---|---|---|---|
| 057OGa | Segelclub Rhe, Rotherbaum | 53.55912 | 9.99605 | 4 | 3 | 5G |
| 059OGa | Uni Segelclub, Uhlenhorst | 53.57696 | 10.01017 | 5 | 3 | 5G |
| 060OGa | Uni Rudersteg, Eppendorf | 53.58521 | 9.99019 | 4 | 3 | 5G |
| 061OGa | Bootssteg Bobby Reich, Winterhude | 53.57979 | 10.00210 | 3 | 3 | 5G |
| 063OGa | Immanuel-Kant Gymnasium, Marmstorf | 53.43041 | 9.96692 | 41 | 3 | 6 |
| 066OGa | Tennisclub Eichtalpark, Wandsbek | 53.58045 | 10.09769 | 13 | 3 | 6A |
| 068OGa[1] | Billwerder Billdeich, Oberbillwerder | 53.49763 | 10.17424 | 2 | 3 | 9 |
| 069OGa | Europaring, Winsen | 53.35427 | 10.20215 | 6 | 3 | 6 |
| 070PGa | Ilmenaudeich, Tönnhausen | 53.37063 | 10.26575 | 4 | 3 | 6 |
| 071PGa | Laßrönner Dorfstraße, Laßrönne | 53.38721 | 10.23099 | 4 | 3 | 6 |
| 072PGa | Hoopter Straße, Stöckte | 53.37869 | 10.19198 | 3 | 3 | 6 |
| 073PGa | Im Schönenfelde, Wilhelmsburg | 53.49830 | 10.01601 | 2 | 3 | 6 |
| 074PGa | Am Eisenwerk, Jarrestadt | 53.58654 | 10.03488 | 8 | 3 | 5 |
| 075PGa | Himmelshorst, Großhansdorf | 53.65183 | 10.29073 | 49 | 3 | 6 |
| 076PGa | Gartenstraße, Trittau | 53.62002 | 10.39891 | 42 | 2 | 6 |
| 077PGa | Hamburger Straße, Ahrensburg | 53.66481 | 10.22399 | 47 | 3 | 6 |
| 078PGa | Waldweg, Immenbeck | 53.44932 | 9.73072 | 22 | 2 | 6 |
| 079PGa | Sumpfveilchenweg, Langenhorn | 53.67367 | 10.03292 | 32 | 3 | 6 |
| 080PGa | Lupinenkamp, Fuhlsbüttel | 53.63517 | 10.02261 | 22 | 3 | 6 |
| 082PGa | Ochsenwerder Norderdeich, Ochsenwerder | 53.46314 | 10.11983 | 2 | 3 | 9 |
| 083PGa[2] | Norderquerweg, Kirchwerder | 53.42645 | 10.19333 | 2 | 3 | D |
| 084PGa | Overdeich, Over | 53.43314 | 10.09711 | 3 | 3 | 6 |
| 085PGa | Jahnstraße, Ashausen | 53.36222 | 10.12728 | 21 | 3 | 6 |
| 086PGa | Kleingartenverein Wilstorf | 53.44632 | 9.98363 | 28 | 3 | 6 |
| 087PGa | Brookwisch, Iserbrook | 53.57825 | 9.82787 | 26 | 3 | 6 |
| 088PGa | Achter de Höf, Rissen | 53.57922 | 9.75940 | 23 | 3 | 6 |
| 089PGa | Kleingartenverein Stellingen | 53.59163 | 9.94172 | 18 | 3 | 6 |
| 090PGa | Kleingartenverein Lokstedt | 53.58656 | 9.95912 | 14 | 3 | 6 |
| 091PGa | Fährstraße, Wilhelmsburg | 53.51829 | 9.98068 | 4 | 3 | 5 |
| 092PGa | Holländische Reihe, Ottensen | 53.54770 | 9.93032 | 31 | 15 | 2 |

[1] APOLLO station replaced on 25 June 2020

[2] APOLLO station replaced by WXT station on 25 June 2020

| Identifier | Name | Latitude (°N) | Longitude (°E) | Altitude (m) | Sensor height (m) | LCZ |
|---|---|---|---|---|---|---|
| 093PGa | Kleingartenverein Niendorfer Gehege | 53.61008 | 9.91888 | 12 | 3 | 6 |
| 094PGa | Harzburger Weg, Niendorf | 53.63124 | 9.94628 | 21 | 3 | 6 |
| 095PGa | Rantzaustraße, Ahrensburg | 53.66977 | 10.21542 | 47 | 3 | 6 |
| 096PGa | Godenwind, Mümmelmannsberg | 53.52577 | 10.14109 | 30 | 3 | 6 |
| 097PGa | Parchimer Straße, Rahlstedt | 53.60891 | 10.16041 | 30 | 3 | 6 |
| 098PGa | Musilweg, Wilstorf | 53.44273 | 9.99474 | 27 | 3 | 6 |
| 099PGa | Schlossgarten, Wandsbek | 53.57030 | 10.06929 | 13 | 3 | 6 |
| 100PGa | Hamfelderedder, Börnsen | 53.47537 | 10.28655 | 55 | 3 | 6 |
| 102PGa | Abelke-Bleken-Ring, Ochsenwerder | 53.47363 | 10.08330 | 1 | 3 | 6 |
| 103PGa | Burbekstraße, Eidelstedt | 53.61903 | 9.87959 | 22 | 3 | 6 |
| 104PGa | Luxweg, Billwerder | 53.49826 | 10.12280 | 1 | 3 | 6 |
| 105PGa | Barsbütteler Landstraße, Barsbüttel | 53.56667 | 10.19619 | 37 | 3 | 6 |
| 106PGa | Kleingartenverein Allermöhe | 53.48406 | 10.14210 | 2 | 3 | 6 |
| 107PGa | Ansorgestraße, Othmarschen | 53.55108 | 9.89326 | 23 | 3 | 6 |
| 108PGa | Auf dem Königslande, Wandsbek | 53.58045 | 10.07517 | 15 | 3 | 5 |
| 109PGa | Eenstock, Bramfeld | 53.60457 | 10.09405 | 18 | 3 | 6 |
| 111OGa | Suntrace, Altona | 53.54436 | 9.93413 | 5 | 10 | 5G |
| 112PGa | Teubnerweg, Billstedt | 53.53520 | 10.12454 | 17 | 3 | 6 |
| 113PGa | Steinbeker Straße, Hamm | 53.54563 | 10.05597 | 6 | 7 | 5G |

**Table 4.** As Table 3, but for WXT measurement sites.

| Identifier | Name | Latitude (°N) | Longitude (°E) | Altitude (m) | Sensor height (m) | LCZ |
|---|---|---|---|---|---|---|
| 002MIw | Wettermast, Billwerder | 53.51996 | 10.10515 | 1 | 3 | 6D |
| 004MIw | Wasserwerk Baursberg, Blankenese | 53.56692 | 9.78784 | 82 | 2 | 6 |
| 005WSw | Wetterstation Langenhorn Grünfläche | 53.67731 | 10.02480 | 24 | 3 | 6B |
| 006WSw | Wetterstation Stellingen Grünfläche | 53.60056 | 9.93568 | 14 | 3 | 6B |
| 017LMw | Luftmessnetz Finkenwerder | 53.53621 | 9.84419 | 6 | 5 | 6 |
| 035KBw | Gut Karlshöhe, Bramfeld | 53.63052 | 10.11114 | 31 | 3 | 6 |
| 044OGw | Fährhaus Tatenberg | 53.49537 | 10.08528 | 2 | 3 | 9 |
| 045OGw | Windpark Curslack | 53.47102 | 10.20700 | 2 | 3 | D |
| 047OGw | Bogenwiese Holm | 53.61872 | 9.68062 | 15 | 3 | 6 |
| 048OGw[1] | Segelflugplatz Boberg | 53.51360 | 10.15142 | 2 | 3 | 9 |
| 050OGw | Bogenschießplatz Stöckte | 53.37930 | 10.18809 | 1 | 3 | 6D |
| 055OGw | Obsthof Schuback, Mittelnkirchen | 53.53679 | 9.64878 | 1 | 3 | B |
| 062OGw | Segelverein Neuländer See | 53.45972 | 10.01936 | 1 | 3 | 9G |
| 064OGw | Biohof Obermeyer, Emmelndorf | 53.40151 | 9.96742 | 49 | 3 | 9 |
| 065OGw | Greenpeace, Wilhelmsburg | 53.50515 | 9.96698 | 8 | 3 | 10G |
| 067OGw | DWD Flughafen, Fuhlsbüttel | 53.63327 | 9.98796 | 9 | 3 | 9 |
| 081PGw | Süderquerweg, Kirchwerder | 53.42561 | 10.13946 | 3 | 3 | 6 |
| 083PGw | Norderquerweg, Kirchwerder | 53.42645 | 10.19333 | 2 | 3 | D |
| 101PGw | Reinbeker Weg, Brunsbek | 53.59054 | 10.26513 | 58 | 3 | 6 |
| 110OGw | Öjendorfer See, Billstedt | 53.55152 | 10.13930 | 13 | 3 | B |
| 114UHw[2] | Geomatikum, Dachterasse 18. Stock | 53.56819 | 9.97483 | 16 | 72 | 4 |

[1]WXT536 sensor replaced on 31 July 2020

[2]WXT536 sensor replaced on 10 June 2020