# Peer review of "Sub-mesoscale observations of convective cold pools with a dense station network in Hamburg, Germany"

_Earth System Science Data, 2021_

## Author Response (AR1)

**Author's response to reviews on ESSD-2021-424**

The authors are thankful to the editor for coordinating the review process and to the two referees for taking their time to carefully reviewing the manuscript and providing detailed and helpful comments to improve the quality of the manuscript. We have attached the revised version of the manuscript. We hope to have adequately addressed all of the concerns raised by the referees. In case of disagreement with individual comments, we have expressed our reasoning. Please find the details replies (in green) to the referee comments along with the line numbers of relevant changes to the manuscript (referring to the revised version; in orange) in the following.

Some additional notes on the revision of the data set and manuscript:

- The DOI of the data set was updated since the data licence was changed from CC BY-SA 4.0 to CC BY 4.0 as requested by ESSD. The version number was not increased since the data itself are unchanged.
- We have incorrectly cited the threshold for cold pool detection at the beginning of section 6.1 (L276). It is 2 K in 20 min instead of 30 min.
- We have added a reference to X-band radar data (Burgemeister et al., 2022) used for plausibility checks for the identification of cold pool events. Currently, the reference only points to meta data but not the data set itself. The DOI of the data will be available soon and we will add it to the citation during the type-setting process in case of acceptance.
- We have adapted the short summary to the wider focus of the manuscript reflected by the new title.

Bastian Kirsch (University of Hamburg), on behalf of all co-authors
04 July 2022

**Review 1**

summary. This manuscript describes the dataset, including instrumentation and network deployment, associated with the FESST@HH experiment designed to study the sub-mesocale structure of convective cold pools. The authors also present some preliminary analysis of a cold pool event, nocturnal urban heat island, and turbulent temperature fluctuations. The manuscript is well-written, dataset well-described, and the topic relevant to the journal. Some comments are suggested for consideration.

Comments

1. The title does not quite represent the manuscript content given that it is not meant to be a detailed analysis of the observed cold pools' sub-mesoscale structure, but rather to describe the dataset and present some preliminary findings on several different topics not only related to cold pools. Re-consideration of the title is suggested.
   We agree and will change the title to "Sub-mesoscale meteorological observations with a dense station network during FESST@HH 2020". (Title)

2. L46-48: These statements are not actually correct, re-wording is suggested based on the C3LOUD-Ex reference.
   We will rephrase the statement based on the suggestion of reviewer 2, who was involved in C3LOUD-Ex. (L46-51)

3. L95-99 and Fig 3: The APOLLO sensor is smoother overall than the Ultrasonic measurement, which appears to capture higher magnitudes of variability, and the APOLLO sensor's running mean does not always match the running mean of the Ultrasonic sensor (e.g. near minutes 3-4, 7-7.5, 9-10). What could cause these differences? The corresponding text (e.g. L97) seems to skate over the differences that are seen in Fig 3.
   The APOLLO data shown in Fig. 3 are un-smoothed raw measurements, whereas the ultrasonic sensor data are 1-s averages of the original 20-Hz data. Still, the ultrasonic sensor adapts instantaneously to sub-second scale fluctuations in temperature due to its measurement principle, whereas the response time (or e-folding time constant) of the APOLLO sensor is on the order of seconds. This explains the differences between both sensors, however, the point we want to make is that the APOLLO sensor captures the shape and strength of the temperature drop without any apparent lag with respect to the inertia-free sensor, meaning that its response time is virtually negligible for measuring cold pools. We will modify the sentence for more clarity. (L101-105)

4. Section 3.1: Does the coverage of sites permit study of land-atmosphere interactions for different land classes? This could be another interesting application of the dataset.
   Yes, this idea would fit to the preliminary analyses shown in section 6.3. One caveat is of course the lack of measurements of turbulent fluxes and/or surface properties (temperature, soil moisture), which need to be related to the discussed temperature fluctuations. Measurements of surface temperature and soil moisture were added in a follow-up experiment. We will mention this in the revised version. (L362-364)

5. L247, section 6.1: Can weaker or dissipating cold pools be detected with these data? The criteria of -2K seems somewhat restrictive?
Yes, for more detailed studies this threshold can be adapted, however, the rather conservative threshold of -2 K has proven to be useful to discriminate the spatial cold pool signal from other source of variability within the station network. We will reformulate the respective part to clarify this. (L276-277)

6. L254 "deepened" – Re-wording is suggested here, since the dataset does not contain observations of cold pool depth.
Thanks, we will do so. (L291)

7. L273: How can there be an expected range of propagation velocities without any observations of cold pool depth? Some additional explanation would be helpful in this section.
We agree that the term "expected range" is misleading in this context. We will rephrase the sentence accordingly. (L310-313)

8. L276-282: The authors could consider dynamic contributions to pressure perturbations in this discussion. The fluctuations in both temperature and pressure for 104PGa shown in Figure 9 are intriguing. Additionally, it is difficult to relate pressure perturbations to the strength of surface-observed cooling because pressure perturbations are also related to the depth of cooling as well as the dynamic contributions. It has been well-known in the literature that pressure perturbations in cold pools are controlled by multiple factors aside from just hydrostatic cooling.
Yes, this is true. Our wording was not clear enough on this topic. We will specifically mentioning the dynamic effects in the revised version. (L320)

9. Figure 10: It would be helpful to indicate the urban and rural stations somehow in panel a, such as by different line styles or thickness.
We believe that the message of the plot, the systematic difference in diurnal cycle between urban and rural stations, becomes clearer when only highlighting selected locations that are representative for the two regimes. Indeed, we missed to clearly indicate which of the two highlighted stations refer to which regime. We will revise the figure caption accordingly. However, a more detailed analysis of the spectrum between the two extremes is out of scope of this study. (Caption Fig. 11)

**Review 2**

Review of essd-2021-424
Aryeh Jacob Drager, University of Nebraska–Lincoln, aryeh.drager16@alumni.colostate.edu

**Overall:**

This manuscript introduces a data set associated with a dense network of weather stations making observations during the June – August 2020 FESST@HH field experiment. The goal of the field experiment was to shed light on sub-mesoscale features of convective cold pools, and the data also seem to produce interesting insights about urban climate effects (although this is not my area of expertise). Despite the excessive length of my review, I should emphasize that the manuscript and data set are generally well constructed. However, the data set and associated manuscript contain several limitations that I would like to see addressed before I can recommend acceptance.

My review is organized into a section on the manuscript and a section on the data set. Each section is organized primarily chronologically. In the manuscript section, I have separated the substantive questions and comments from the minor technical corrections. The data set section is organized roughly into requests for additional data and comments on existing data. I have also uploaded some Matlab code and results associated with sample analyses that I completed.

**Manuscript**

**Substantive Comments on the Manuscript:**

**Line 18:** Please explain the full field experiment acronym. As I read the manuscript, I found myself wondering why *H* appeared in the acronym twice. I did not realize until I saw the 2 May 2022 version of the data set that "@HH" meant "at Home, Hamburg" (or something similar, I think).
You are right, we missed to explain that. "HH" is Hamburg's license plate prefix, meaning "Hansestadt Hamburg" (=Hanseatic City of Hamburg). We will add this to the explanation of the acronym. (L3,17)

**Line 33:** All of these references examine oceanic convection exclusively. Drager et al. (2020) discuss these triggering mechanisms in a continental environment, and I recommend citing this article here.
Thanks, we will add your paper here. (L34)

Drager, A. J., Grant, L. D., & van den Heever, S. C. (2020). Cold pool responses to changes in soil moisture. Journal of Advances in Modeling Earth Systems, 12, e2019MS001922. https://doi.org/10.1029/2019MS001922

**Line 34:** There are many earlier references regarding cold pool influences on the organization of convective clouds that should be cited here. These include:
In the interest of text length, we will just add the RKW 1988 citation. (L35)

Purdom, J. F. W. (1976). Some uses of high-resolution GOES imagery in the mesoscale forecasting of convection and its behavior, Mon. Weather Rev., 104(12), 1474–1483, https://doi.org/10.1175/1520-0493(1976)104%3C1474:SUOHRG%3E2.0.CO;2

Purdom, J. F. W. (1982). Subjective interpretation of geostationary satellite data for nowcasting. In K. Browning (Ed.), Nowcasting (pp. 149–166). London: Academic Press.

Rotunno, R., Klemp, J. B., & Weisman, M. L. (1988). A theory for strong, long-lived squall lines. *Journal of the Atmospheric Sciences*, 45(3), 463– 485. https://doi.org/10.1175/1520-0469(1988)045%3C0463:ATFSLL%3E2.0.CO;2

Wilson, J. W*., & Schreiber, W. E. (1986)*. Initiation of convective storms at radar-observed boundary-layer convergence lines*. Monthly Weather Review, 114(12), 2516–2536*. https://doi.org/10.1175/1520-0493(1986)114<2516:IOCSAR>2.0.CO;2

**Line 38:** I'm not sure what "Daw" refers to (there is no year provided, and there is no corresponding entry in the References list), and in any case, some earlier references should be provided here since Li et al. (2015) was certainly not the first to discuss this topic. A couple of suggested references are below.
Thanks for the suggestion. We will fix the issue with the Dawson et al, 2010 citation and add the Smagorinsky citation. (L39)

Deardorff, J. W. (1980). Stratocumulus-capped mixed layers derived from a three-dimensional model. Boundary-Layer Meteorology, 18(4), 495–527. https://doi.org/10.1007/BF00119502

Smagorinsky, J. (1963)*, General circulation experiments with the primitive equations. I. The basic experiment, Mon. Weather Rev., 91(3), 99–164,* https://doi.org/10.1175/1520-0493(1963)091<0099:GCEWTP>2.3.CO;2

**Lines 45-48:** As someone who participated in the C$^3$LOUD-Ex field campaign, I take issue with some aspects of the way it is characterized here. Firstly, "convective updrafts and downdrafts" is not quite correct. We were interested in characterizing convective updrafts and cold pools (the "O" in C$^3$LOUD-Ex stands for "Outflows," i.e., cold pools). Moreover, although there were indeed only three surface stations, the surface stations were portable so as to allow for targeted observations of cold pools. The line of surface stations (with quasi-stationary UAS hovering above each surface station, along with radiosonde launches, arranged in a coplanar "flying curtain") was always oriented parallel to the advancing gust front, such that a time-to-space conversion would be able to characterize the three-dimensional structure of the cold pool as the gust front passed through the flying curtain. There certainly exist limitations to this

approach, and I do not doubt that more surface stations would have been useful. However, the comparison of the targeted observations of C³LOUD-Ex to the stationary surface station network of FESST@HH is not quite an apples-to-apples comparison. I would request the following rewording in order to clarify more precisely the contrast between C³LOUD-Ex's scope and FESST@HH's scope:

In a recent study, van den Heever et al. (2021) described the use of uncrewed aerial systems, regional radars, radiosondes, and surface stations during the C³LOUD-Ex campaign to characterize convective updrafts and cold pools on scales between $O(100)$ m and $O(1)$ km. Their portable "Flying Curtain" sensing network collected in-situ observations of cold pools over three closely-spaced geographical locations and revealed that cold pool temperatures can exhibit variability on spatial scales of $O(100)$ m as well as $O(1)$ km. However, given the small geographical extent of the in-situ network, C³LOUD-Ex was unable to characterize the complete surface temperature structure of any individual cold pool.

Thank you for the clarification. We will include your suggested wording into the revised version. (L46-51)

**Line 58:** In line 246, you mention that there were 37 cold pool events during the three-month measurement period, which translates to ~12 cold pool events per month. This seems to be significantly higher than the expected amount (7 cold pool events per month). Was 2020 a lucky year for cold pools, or is the discrepancy due to the greater area covered by the FESST@HH domain, as compared to the single location examined in Kirsch et al., 2021a?

Yes, we see the greater area compared to the point measurement as one reason for the higher number. Another explanation is that Kirsch et al., 2021a only include cold pool events that produce both a temperature drop and measurable rainfall at the observation point. For the identification of the 37 cold pools events we only applied the temperature criterion and checked the plausibility based on the visual inspection of radar imagery to include cold pools propagating away from the parent rainfall or without any surface rainfall. This approach was not feasible for the multi-year data set in Kirsch et al., 2021a. We will include this explanation in the first paragraph of section 6.1. (L279-281)

**Section 2.1:** A major limitation of this data set is the APOLLO stations' lack of humidity and wind measurements. Wind is an essential metric for identifying cold pool boundaries (e.g., Fournier and Haerter, 2019, citation below). Additionally, as is discussed by Kirsch et al., 2021a, there is still much about the near-surface moisture structure of cold pools (e.g., moisture rings) that is poorly understood. Could you discuss why the decision was made not to include these measurements?

The implementation of a wind sensor would have considerably restricted the choice of potential measurement locations due to the influence of nearby obstacles. This is not the case for temperature and pressure. A humidity sensor was not included since a precise measurement would have considerably increased the technical effort and financial costs per station. Additionally, humidity has not proven to be a reliable quantity for detecting real (i.e., not simulated) cold pools over land. Still, the WXT stations help to characterize the humidity

and wind field during the cold pool events, albeit at fewer locations. We will add a short discussion on this topic at the end of section 2.1. (L108-110)

Fournier, M. B., & Haerter, J. O. (2019). Tracking the gust fronts of convective cold pools. Journal of Geophysical Research: Atmospheres, 124, 11,103 – 11,117. https://doi.org/10.1029/2019JD030980

**Sections 2.1 and 2.2:** I am wondering what impact the lack of sensor aspiration may have had on the results obtained during the field experiment.
As cold pools are usually associated with wind speed increases of >= 3 m/s (the absolute wind speed is higher), the sensors are sufficiently ventilated during cold pools passages. Therefore, we don't expect a significant impact. We will add a corresponding sentence to the last paragraph of section 2.1 (L101). We will also add the term "passively ventilated" in the first paragraph of section 2.1 to be clear on this. (L81)

**Lines 79-80:** A few questions here:
- Is this an *e*-folding time constant [i.e., time to complete $100 \times (1 − 1/e) \approx 63.2\%$ of the response to a step-function increase or decrease], or is it some other metric? Yes, it is exactly that. (L98-99)
- What procedures were used in obtaining this estimated time constant? "Wind tunnel laboratory tests" is not a sufficient description. In the wind tunnel laboratory experiment we have heated the NTC temperature sensor including its housing to about 50 °C and let it cool down to room temperature under different ventilation conditions. (L98-100)
- How does the response time vary as a function of wind speed? As expected, the response time decreases with increasing wind speed.

We will move the respective part to the last paragraph of section 2.1 and elaborate on the requested details. We will also add the corresponding details for the WXT stations. (L127-128)

**Line 94:** Do you have plans to draft instructions for acquiring the required parts and building the stations? Is the €300 figure still accurate? What are the labor costs for assembling the stations?
No, we don't have such plans but we would be happy if the stations would be re-used in future projects. The 300 € is still accurate. The labor costs are in fact difficult to assess. The development and assembling took about three years and involved (part-time) work of one engineer, one PhD student and two student workers.

**Lines 95-97:** A few questions here:
- Could you provide more details regarding the comparison with the "inertia-free ultrasonic sensor"? For example, what type of ultrasonic sensor was used? It was an ultrasonic anemometer METEK uSonic-3 Scientific. (Caption Fig. 3)
- What does "almost instantaneously" mean – can you quantify this? No, we cannot quantify this, but we qualitatively see that the sensor captures the minute-scale shape

of the signal without any apparent lag and even represents second-scale fluctuations observed by the inertia-free ultrasonic sensor. (L103-105)

- What does "high-frequency" mean – can you quantify this? I'm not sure whether I should be focusing on the oscillations in Figure 3 that last for a few seconds or those that last for about a minute.

In this context we refer to the second-scale fluctuations.

We will rephrase the sentence for clarity. (L103-105)

**Lines 98-99:** A few questions here:
- What does "considerable" mean in this context – can you quantify the impact?
- How was this assessed? I am interested to see the "not shown" data and learn more about how they were collected.
- Was this assed under a variety of cold pool conditions, including clear versus precipitating conditions?

No, we cannot quantify the impact. We did a 10-day test experiment in the field and placed the APOLLOs in different environments, like on a grass field, in a forest and near buildings and asphalt surfaces. We were actually lucky enough to capture two cold pool events. Apart from differences in signal before and after the temperature drop, we do not see systematic impacts on the amplitude or shape of the cold pool signal (see plot below). This is also what we would a priori expect for temperature as the initial strong drop in temperature is mostly determined by the atmosphere via the downdraft air falling down on the surface. The effect of the surface, if anything, would be more visible for the recovery phase, or for cold pools without precipitation. Still, we do not believe that these results are substantial enough to be extensively discussed here. We will rephrase the sentence to be more precise on this issue. (L105-108)

[Figure]

**Section 2.2:** What impacts do obstacles and surface properties have on the cold pool signal in the WXT data? This is not discussed.

Thanks for the suggestion. As discussed in the previous answer, we don't expect a systematic impact of the environment on the temperature, pressure and humidity signal during cold pools. This is of course not true for the wind, which is affected by near obstacles and vegetation that may heavily alter wind speed and wind direction. Therefore, the WXT stations were preferably set up at locations with few obstacles that could disturb the near-surface flow field. We will add a short paragraph on this issue at the end of the section (L134-138) and will also briefly mention it in section 3.1 (L152-153). In this context, we will also add a reference to the site photographs attached to the data set at the end of section 4.3 (L246-247), which we missed to include in the first version of the manuscript and may help to interpret the wind measurements of individual stations.

**Line 136:** What is the significance of the number following each ± symbol? Are these standard deviations? The full ranges? Please make this explicit.

It is the standard deviation. We will add it accordingly. (L154)

**Line 148:** What is meant by "near" and "large"? Can you either quantify these or provide a map showing these particular bodies of water?

We will specify "near" as "< 50 m away" in the text and drop "large" since we specify the water bodies in the following. (L167-168)

**Lines 150-151:** Can you provide exact dates? I do not see any data from early September in the data set.

The experiment (=data set) ran from 1 June to 31 August 2020, whereas we started to set up the instruments on 28 May and started to dismantle them on 1 September. We will modify the text here to avoid confusion. (L170-171)

**Line 182:** Does "midnight" refer to local time or UTC?

UTC. This will be clarified. (L201)

**Lines 182-183:** What is a measurement "telegram"? (Note: This word was also used in Line 88.) I am unfamiliar with the use of this word outside of the context of 19$^{th}$-century technology (telegraphs, Morse code, etc.). Can this wording be made more straightforward?

We will replace the term by "message" (line 88) and "data" (line 182). (L90,201)

**Sections 4.1 and 4.2:** Can the level 0 and level 1 data be added to the public data set?

Yes, this would be possible, but we see no added value in publishing these hard-to-use and non-quality-controlled data.

**Lines 205-207:** Two things:
- I am not convinced that 15 K is an appropriate threshold here, given that a cold pool may affect only one station at a given time as it begins to spread across the domain. Cold pool temperature drops of 15 K do not seem implausible.

This threshold is not physically based but does its job in removing all obviously erroneous data but still preserving the cold pools in postprocessing.
- What does it mean that these three stations had erroneous temperature sensors? How is this defined, and how was it determined that these three sensors were erroneous?

These three stations occasionally produced unphysical spikes in the temperature data. We will clarify this in the text. (L225-227)

**Lines 213-214:** These biases could have changed over the course of the measurement campaign, as it is possible for individual sensors to drift by different amounts over time. When did the week-long calibration occur, relative to the time period of the measurement campaign? Ideally I think this calibration should have been done both before and after the measurement campaign, at the very least. How were the WXT sensors calibrated? What about the pressure sensors in the APOLLO stations? The manuscript does not seem to provide any information in this regard.

The APOLLO temperature calibration period took place only before the experiment between January and June 2020. The APOLLO pressure sensors were not calibrated since in test measurements the absolute values showed an oscillating drift pattern with an amplitude and

period of about 0.5 hPa and 10 days. The WXT sensors are calibrated by the manufacturer. We have will add the respective information to the manuscript. (L233-237)

**Section 5.2:** How is a "day" defined/delineated? Is this from midnight to midnight UTC, or is a different definition used (e.g., midnight to midnight local time)?
In UTC. We will add the information to the text and the caption of Fig. 7. (L262, Caption Fig. 7)

**Lines 247-248:** How is "associated with rainfall" determined/defined?
Thanks, we are not clear on this point. We checked the cold pool passages identified by our algorithm based on the visual inspection of rainfall radar data. We will revise this part accordingly. (L277-279)

**Lines 250-251:** How was this information determined? Were radar retrievals used, and if so, can they be shown here?
Yes, this information is based on observations from a X-band rain radar operated by our institute. The plot below (taken from a current study in preparation) shows a corresponding snapshot along with the spatially interpolated temperature and pressure data and wind observations of WXT stations. However, we decided to not include the radar data in the present study to avoid confusion and keep focus on the data set described here. The X-band radar data will be published separately.

[Figure]

**Lines 254-255:** The cold pool appears to extend beyond the eastern edge of the network, so it seems difficult to draw any conclusions regarding the actual size of the cold pool.
This is true. We will add for clarification that we assume a roughly elliptical shape for the cold pool. (L293)

**Line 269:** Is this gradient based on the absolute temperatures or the perturbations? Please clarify.

It refers to the perturbations. We will add this. (L306)

**Lines 273-275:** Can you expand upon how the expected range was determined?

We will rephrase the sentence to avoid the misleading term "expected range". (L310-313)

**Lines 280-282:** Other studies have discussed non-hydrostatic pressure effects associated with downdrafts and cold pools, and they should be cited here. Here's one example, though there are probably some earlier studies as well:

We have will add two related citations here. (L320-321)

Wu, F., & K. Lombardo (2021). Precipitation enhancements in squall lines moving over mountainous coastal regions. Journal of the Atmospheric Sciences, 78(10), 3089–3113. https://doi.org/10.1175/JAS-D-20-0222.1

**Lines 289-290:** What does it mean for an environment to be "weakly sealed"? How is "sealing" quantified? I am not familiar with this concept, and I expect that many (if not most) readers—at least those whose primary interest is cold pools—will not be familiar with it either. Could you provide a brief definition, or at least cite and refer readers to some sources that explain this concept?

We will add the definition "mostly natural" for "weakly sealed". We believe that this information is sufficient to follow our very rough interpretation of urban effects here. (L329)

**Lines 290-291 and Figure 10:** According to https://data.giss.nasa.gov/modelE/ar5plots/srlocat.html, the day length in Hamburg (53.55°N, 10.10°E) changes by 3.32 hours over the course of the interval from 15 June to 31 August (maximum of 17.05 hours around 20 June, minimum of 13.73 hours on 31 August). Daily mean insolation is also changing, with the 24-hour average maximizing at 482.27 W/m$^2$ on 20 June and minimizing at 334.70 W/m$^2$ on 31 August.

With the caveat that I am not an expert on quantifying the urban heat island effect:

Given that the sunset and sunrise timing are changing over the averaging period, I am wondering if it would make more sense to shift and rescale the time axis prior to averaging so that the sunset and sunrise times are aligned. This would preserve the abruptness of any temperature shifts associated with sunset and sunrise, and would minimize "smearing." Alternatively, since the daily sunset and sunrise times do not change very much near the solstice, limiting the averaging period to 15-30 June, and excluding the months of July and August, could have a similar effect.

This remark is perfectly justified; however, a more advanced analysis of the urban heat island effect is out of scope of this paper. The purpose of the paragraph is to showcase the effects and processes captured by the network as a starting point for future and more detailed studies.

**Lines 320-321:** Can you expand on how these additional experiments were conducted, and how this conclusion was drawn?

The experiments show that sigma_T of two co-located APOLLOs is identical when one sensor shield is protected from direct solar radiation. This suggests that the temperature fluctuations are not artificially generated by the heating of the sensor shield. We will rephrase the sentence accordingly to be more specific here. (L359-360)

**Figure 4:** The four lowermost white-text labels ("3-m mast" and below) are displaced far to the left of the items they describe. Please either add arrows or move the labels so that it is obvious what they are referring to.

Thanks, this will be done. We will also also update Fig. 1 for readability. (Fig. 1,4)

**Figure 8:** Please plot the WXT stations using a different shape (perhaps a square) so that they can be distinguished more easily from the APOLLO stations.

We will do so. (Fig. 9)

**Table 1:** What is the relative accuracy of the pressure sensor for temperatures below 25°C? Please provide this information.

We do not have this information, but only the information provided by the manufacturer.

**Table 2:** Can you provide additional wind speed and direction accuracy estimates for weaker winds, given that winds as strong as 10 m/s seem not seem to occur very often here (per Figure 8)?

Same here.

**Minor Technical Comments:**
Spelling, grammar, rewording (including minor alterations in the sentence meaning)

Thank you for the corrections. We will include them in the revised version if not indicated otherwise.

Line 3: sub-mesocale → sub-mesoscale
Lines 9-10: supported the station maintenance → supported station maintenance
Line 10: add a comma after "their measurement characteristics"
Line 12: add a comma after "inside a cold pool"
Line 13: its size → a cold pool's size
Lines 14-15: as an expression of → associated with
Line 18: Conventional meteorological → Conventional mesoscale meteorological
Line 19: essential to obtain → essential in order to obtain
Line 20: techniques help → techniques can help
Line 21: precipitation or wind speed → precipitation and wind speed
Line 23: The information → Information
Line 25: This fact qualifies them as the → Therefore, cold pools are the We will also drop the "therefore" in the previous sentence.

Line 29: several tens of km → hundreds of km (note: squall lines can extend across hundreds of km)

Line 30: temperature falling by more than 10 K → temperature decreases that can exceed 10 K

Line 30-31: propagates away from → propagates horizontally away from

Line 36: Grant and Heever → Grant and van den Heever

Line 42: aircrafts → aircraft

Line 49: the the → the

Line 52: aim → idea

Line 56: cheap → inexpensive (Note: "cheap" can carry connotations of poor quality)

Line 60: allows to study → allows for the assessment of

Line 62: in response to → due to

Line 68: cold pool fronts → cold pool gust fronts We will keep our wording here because "gust front" is more associated with wind instead of temperature, which is the key variable for our analysis

Line 69: had to operate independent of → had to be able to operate independently of

Lines 69-70: to facilitate the search for appropriate site locations → given site location constraints

Line 70: According to → Based on

Lines 71-72: WXT weather stations based → WXT weather stations (see Section 2.2) based

Line 73: APOLLO → APOLLO stations

Line 76: source and → source, and

Line 82: whereas → and

Line 84: written → recorded

Line 87: to correct → correction of

Line 87: drifting → drift

Line 100: WXT weather station → WXT stations We will just add a "s".

Line 104: direction and precipitation → direction, and precipitation

Line 112: rains drops → rain drops

Line 118: The units of the station are powered by → Each station is powered by

Line 120: sun light → sunlight

Line 126: in response to → associated with

Line 128: home office → pandemic-related

Line 133: randomly → non-uniformly

Line 133: the tendency for → generally

Line 134: The random arrangement results → The arrangement of stations results

Line 139: altitude of all measurement site lies between → altitudes of all measurement sites lie between

Line 141: impact → impacts

Line 148: 10 → Ten (Note: this may depend on ESSD's editorial style)

Line 152: permissions at short notice for using → permission at short notice to use

Line 165: home office not only affected → pandemic-related restrictions affected not only

Line 169: the data and → the data, and

Line 169: public ground → public grounds

Line 175: In contrast → By contrast

Line 190: regular → standardized We think "regular" is more precise here.

Line 206: sensor → sensors

Line 207: applied and → applied, and

Line 210: near by → nearby

Line 212: sensor; → sensor; and

Line 214: (114UHw); → (114UHw).

Line 218: meta data → metadata

Line 219: above ground and → above ground; and We will add a comma instead.

Line 229: higher stability of the power supply → greater stability of the WXT power supply

Line 229: generally higher availability → generally greater availability

Line 235: more than → maximum daily temperatures exceeding

Line 236: by a phase of relatively cold temperatures with → by relatively cold temperatures, with

Line 237: Rainfall → Measurable rainfall

Line 238: on half of → on approximately half of **OR** on exactly half of (Note: exactly half would be 46 days)

Lines 239-240: June when a strong convective event occurred → June in association with a strong convective event

Line 244: further → additional

Lines 251-252: surface-layer air masses → surface-layer air

Lines 254-255: cold pool deepened to → cold pool temperature perturbation strengthened to

Line 258: early stages of its life → early stages of the cold pool's life

Line 259: cold air masses during → cold-air region during

Line 262: cold pool origin experienced → cold pool center experienced

Line 263: reached its maximum → reached a maximum

Line 266-267: the maximum cooling decreases to about one-third away from the center → the maximum amount of cooling decreases by about two-thirds away from the center

Line 270: (2021) who → (2021), who

Line 270: cold pool properties on scales between $O$(100) m and $O$(1) km → cold pool temperatures of order 1–2 K on scales between $O$(100) m and $O$(1) km We will also add "perturbation".

Line 284: effects as → effects, as

Line 285: surrounding → surroundings

Line 305: by the local → by local

Line 306: condition like → conditions like

Line 312: effectively reduced → effectively reduces

Line 334: their ground as → their grounds as

Lines 335-336: they were never used → they had never been used

Line 340: urban meteorology as the → urban meteorology, as the

Line 347: glass for sub-mesoscale → glass for revealing sub-mesoscale

Line 354: prove of concept → proof of concept

Line 362: analyses helped to → analyses. FA also helped to

Line 363: conflict of interests → conflicts of interest

Line 388: this entry is missing the author(s) and year

Line 416: and Heever, S. C. v. d. → and van den Heever, S. C.
Line 416: Geohphys. → Geophys.
Line 453: Szoeke, S. P. D. → de Szoeke, S. P.
Figure 3 caption: by co-located → by a co-located
Figure 5 caption: sub-class is considered → sub-class is depicted
Figure 7 caption: 5%-quantile and 95%-quantile → 5th percentile and 95th percentile
Figure 8 caption: perturbation smaller than → perturbation stronger than
Figure 8 caption: Length of reference arrow refers to → The length of the reference arrow corresponds to
Table 3 caption: Meta data of → Information about

**Data Set**

**Requested Additions to the Data Set:**

1. The data set would benefit greatly from the addition of appropriate radar and/or geostationary satellite data. This would enable users to contextualize the cold pool observations relative to the locations of rain shafts and clouds, without having to track down these data elsewhere. If there are licensing or other restrictions preventing the inclusion of such data, then it would be helpful if an instructional guide for locating and downloading such data could be included.

This is of course a good idea, but not the scope of the data set. The strengths of the data set are that it is simple, small, and easy to use. We are only giving suggestions of possible implications and applications of the data rather than promoting specific ideas for future studies or other data sets.

2. Please add a list of the 37 cold pool events, either as a table in the manuscript or as part of the data set.

We will add a plot of all cold pools events at the beginning of section 6.1 along with a short discussion on the variability of their signal strength to motivate the following case study. (L281-284, Fig. 8)

3. Please add some derived variables to the data set, such as:
   a. u- and v-components of the horizontal wind at WXT stations
   b. water vapor mixing ratio and/or specific humidity at WXT stations
   c. potential temperature at APOLLO and WXT stations
   d. equivalent potential temperature at WXT stations
   e. virtual potential temperature at WXT stations

All of these quantities can be easily calculated from the available data and in view of disk space we do not want to store redundant information.

**Specific Comments on the Data Set**
*Note: Except where stated otherwise, I analyzed the following version of the data set:*
*https://www.fdr.uni-hamburg.de/record/8973*

4. When I unzipped the main data files using the Finder app on my MacBook (as opposed to, e.g., using a command-line interface), the default names of the directories were not helpful. For instance, for the version of the data set uploaded 2 May 2022, the files "fessthh_uhh_apollo_l2_202006.zip" and "fessthh_uhh_wxt_l2_202006.zip" both unzip to create directories named "fesst-at-home". I realize that these can be altered quite easily; it's just a minor inconvenience.

We are sorry for this inconvenience. However, we would like to refrain from correcting this since an update of the files would be associated with a new DOI for the data set.

5.  → This seems to have been fixed in the version of the data set uploaded 2 May 2022. (However, the "Last updated: 17 March 2021" line at the bottom of the document was not updated!) Only the data licence was updated, not the data itself or the readme file.

6.  → This seems to have been fixed in the version of the data set uploaded 2 May 2022. Yes, well spotted. We took the opportunity to fix this minor issue in the updated data set.

7. The number of seconds since 1 January 1970 at 00:00 UTC does not appear take leap seconds into consideration. This causes an offset of 27 seconds when I analyze the data using Matlab's datetime capabilities.
Yes, this is true. We will add a clarification at the end of section 4.3. (L241-242)

8. The long name and comment for zsl seem to be contradictory. Why not just say "altitude of ground level above mean sea level" or similar?
We take the point that the two information appear confusing at first glance. However, we want to be consistent with the naming of "latitude/longitude of instrument location" and the comment only specifies that this altitude refers to the ground level in contrast to the zag variable. As the name "comment" says, it is not essential to understand the variable but adds precision if needed.

*Details:*

From ncinfo (in Matlab) for fessthh_uhh_apollo_l2_ta_v00_20200810000000.nc:

```
zsl
    Size:      82x1
    Dimensions: station
    Datatype:  single
    Attributes:
         standard_name = 'altitude'
         long_name    = 'altitude of instrument location above mean sea level'
         comment      = 'altitude of ground level'
         units        = 'm'
```

9. Wind direction: How are the degree values handled when wind is from the north, oscillating between 0° and 360°, given that these are averages of the 4-Hz data? Are there any cases where combinations of slightly east of north and slightly west of north average out to 180° (for example)? Or is this done by averaging the u- and v-components over time?
We are not aware of such an issue. In the figure below, we show an example period with frequent oscillation of the wind direction between 0° and 360°. The time series (upper plot) shows the frequent oscillation but the histogram (lower plot) does not show any recorded

values around 180°. This suggests that the internal averaging is correctly implemented by the manufacturer of the WXT sensor.

[Figure]

[Figure]

**10.** Wind gust: I am concerned because of the large gust length (3 seconds) relative to the time interval (10 seconds). What if there's a stronger gust that straddles two time intervals? I think it might make more sense to calculate such wind gusts on a rolling basis or something like that.

The 3-sec gust is indeed internally updated every second. The wind gust value given in the data set is the largest 3-sec gust in the 10-sec interval. We will clarify this in section 2.2. (L120-121)

**11.** Number of characters allocated for station ID, LCZ, etc., as well as rows vs. columns:

Why are 40 characters allocated for each station ID, when each station ID is only 6 characters long? Also, the names are arranged in columns, rather than in rows, which makes them difficult to read in their raw format. This is what it looks like when I read the data into Matlab:

From ncinfo for fessthh_uhh_apollo_l2_pa_v00_20200810000000.nc:

station_id
    Size:     40x82

Dimensions: character,station
Datatype:   char
Attributes:
        long_name = 'station identifier code'

```
  station_id_apollo  ✕
ch 40x82 char

val =

   '0000000000000000000000000000000000000000000000000000000000000000000111111111111'
   '0011111122222222223333333444445555556666667777777777788888888889999999990000000001111'
   '7801568901234567890136790136912347901368901234567890234567890123456789023456789123'
   'WWWWLLLLLLLLLLLKKKKKKKUUUUU00000000000000PPPPPPPPPPPPPPPPPPPPPPPPPPPPPPPPPPPPPPPPPOPP'
   'SSSSMMMMMMMMMMBBBBBBBHHHHHGGGGGGGGGGGGGGGGGGGGGGGGGGGGGGGGGGGGGGGGGGGGGGGGGGGGGGGGG'
   'aaaaaaaaaaaaaaaaaaaaaaaaaaaaaaaaaaaaaaaaaaaaaaaaaaaaaaaaaaaaaaaaaaaaaaaaaaaaaaaaaa'
   '                                                                                  '
   '                                                                                  '
```

(not pictured: 32 additional empty rows)

The same is true for the LCZ information. This would have been more helpful if it had been arranged in rows, rather than columns, and as far as I can tell, none of the LCZs are more than 2 characters long (even though 40 characters are allotted). Compliance with the CF convention for NetCDF files required strings to be represented as character arrays. Therefore, the extra character dimension of length 40 is needed to hold the maximum length of any string variable in the data set, which is the station name (maximum string length = 38).

12. There seems to be an issue with the data at APOLLO station 077PGa, in which data are often missing very close to the XX:00:00 time stamp. The following is a list of "isolated" times for which pressure data are missing at this site. ("Isolated" is defined here to mean that data are only missing for one second at a time. If data are missing for 2 or more consecutive seconds, then the time stamps are not included in the list below.)

| | | |
|---|---|---|
| '10-Jun-2020 18:00:01' | '26-Jun-2020 02:00:16' | '16-Jul-2020 18:00:02' |
| '10-Jun-2020 21:00:08' | '02-Jul-2020 17:00:02' | '16-Jul-2020 23:00:01' |
| '11-Jun-2020 03:00:01' | '03-Jul-2020 00:00:02' | '17-Jul-2020 10:00:11' |
| '11-Jun-2020 06:00:01' | '03-Jul-2020 02:00:02' | '20-Jul-2020 17:00:01' |
| '11-Jun-2020 08:00:33' | '03-Jul-2020 03:00:09' | '22-Jul-2020 01:00:19' |
| '11-Jun-2020 09:00:31' | '03-Jul-2020 08:00:02' | '22-Jul-2020 03:00:01' |
| '14-Jun-2020 09:00:07' | '03-Jul-2020 12:00:14' | '27-Jul-2020 08:00:18' |
| '19-Jun-2020 03:00:25' | '03-Jul-2020 15:00:11' | '27-Jul-2020 08:00:30' |
| '19-Jun-2020 06:59:58' | '06-Jul-2020 02:00:01' | '27-Jul-2020 08:00:48' |
| '19-Jun-2020 09:59:59' | '06-Jul-2020 11:00:01' | '31-Jul-2020 00:00:15' |
| '21-Jun-2020 06:00:02' | '06-Jul-2020 13:00:01' | '01-Aug-2020 00:01:06' |
| '21-Jun-2020 07:00:02' | '07-Jul-2020 02:00:48' | '06-Aug-2020 06:00:02' |
| '21-Jun-2020 09:00:02' | '08-Jul-2020 02:00:01' | '06-Aug-2020 13:00:02' |
| '21-Jun-2020 13:00:14' | '09-Jul-2020 17:00:10' | '06-Aug-2020 14:00:02' |
| '21-Jun-2020 15:00:02' | '16-Jul-2020 11:00:01' | '06-Aug-2020 15:00:02' |

'07-Aug-2020 08:00:00'
'10-Aug-2020 15:00:00'
'11-Aug-2020 20:00:01'
'12-Aug-2020 06:00:42'
'12-Aug-2020 17:00:01'
'15-Aug-2020 19:00:02'
'15-Aug-2020 20:00:14'
'18-Aug-2020 15:00:01'
'21-Aug-2020 22:00:01'
'25-Aug-2020 23:00:00'
'26-Aug-2020 12:00:18'
'26-Aug-2020 12:00:42'
'26-Aug-2020 12:01:01'
'26-Aug-2020 12:01:18'
'26-Aug-2020 12:01:42'
'26-Aug-2020 12:59:59'
'26-Aug-2020 17:02:23'
'26-Aug-2020 18:02:31'
'30-Aug-2020 16:00:01'
'31-Aug-2020 12:00:02'
'31-Aug-2020 16:00:02'
'31-Aug-2020 19:00:02'

I obtained a nearly identical list for temperature data as for pressure data, so I did not include both lists here.

I was curious to see whether anything weird happened when data were not missing, so I plotted 100-second time series of the temperature and pressure centered at 31-Aug-2020 13:00:02, 14:00:02, 15:00:02, 17:00:02, and 18:00:02, and I did not see any anomalous signal in pressure or temperature at the XX:00:02 time stamp.

This issue may exist at other stations, and I will ask the authors to investigate how pervasive the issue is, as well as the potential cause of the missing data. Since the GPS time is logged once per hour, perhaps there is sometimes sufficient clock drift to result in an immediate jump from XX:00:00 to XX:00:02 upon GPS synchronization, resulting in missing data at XX:00:01.

Although this issue seems to result in a very small proportion of missing data, it is inconvenient that the data are missing very close to the XX:00:00 time stamps, as users are more likely to sample the data at/near the "clean" XX:00:00 time stamp than at other, more "random-looking" time stamps like (perhaps) XX:52:23.

I might suggest creating (and documenting) an alternative version of this data set with any one-second gaps filled in with the average of the surrounding values, if this is considered permissible and ethical. Either way, this issue should be documented more fully so that users are aware.

As correctly assumed, this behavior is caused by the time drift correction and the slightly varying activation times of the GPS module. We are aware of this "issue", which is more or less inevitable for this type of observational data. We will add a note on this in section 4.3 of the revised manuscript. (L242-244)

**My own exploration of the data using Matlab R2021a:**

- I was able to reproduce Figure 9 using the data set:

[Figure]

I interpolated data onto a Cartesian grid and plotted the corresponding maps.

[Figure]

The above sample map, which corresponds to Figure 8d in the manuscript, contains some strange spatial oscillations near 53.5°N, 10.15°E. It is unclear to me what is causing these oscillations. It is possible that I have made a mistake somewhere. I have uploaded my code as part of this review, along with the images/animations that were produced for this 10 August 2020 cold pool case-study.

Thank you for the detailed exploration of the data set. We would attribute these spatial oscillations to the linear interpolation that was applied to the irregular distributed station data. A more sophisticated interpolation method (e.g., kriging) and more advanced calculation of the temperature perturbation will produce a smoother field (see plot in our

reply to lines 250/51) that is more tolerant to spatial inconsistencies caused by local temperature fluctuations superposing the cold pool signal. The avoid these issues, that are out of scope of this study, we just included plots of the scattered point measurements.